



# Aerosol-cloud-turbulence interactions in well-coupled Arctic boundary layers over open water

Jan Chylik[1], Dmitry Chechin[2], Regis Dupuy[3], Birte S. Kulla[1], Christof Lüpkes[4], Stephan Mertes[5], Mario Mech[1], and Roel A. J. Neggers[1]

[1]Institute for Geophysics and Meteorology, University of Cologne, Germany.
[2]Obukhov Institute of Atmospheric Physics, Russian Academy of Sciences, Moscow, Russia
[3]Université Clermont Auvergne, CNRS, Laboratoire de Météorologie Physique (LaMP), F-63000 Clermont-Ferrand, France
[4]Alfred Wegener Institute (AWI), Bremerhafen, Germany
[5]Leibniz Institute for Tropospheric Research (TROPOS), Leipzig, Germany

**Correspondence:** Jan Chylik (jchylik@uni-koeln.de)

**Abstract.** Late springtime Arctic mixed-phase convective clouds over open water in the Fram Strait as observed during the recent ACLOUD field campaign are simulated at turbulence-resolving resolutions. The main research objective is to gain more insight into the coupling of these cloud layers to the surface, and into the role played by interactions between aerosol, hydrometeors and turbulence in this process. A composite case is constructed based on data collected by two research aircraft
on 18 June 2017. The boundary conditions and large-scale forcings are based on weather model analyses, yielding a simulation that freely equilibrates towards the observed thermodynamic state. The results are evaluated against a variety of independent aircraft measurements. The observed cloud macro- and microphysical structure is well reproduced, consisting of a stratiform cloud layer in mixed-phase fed by surface-driven convective transport in predominantly liquid phase. Comparison to noseboom turbulence measurements suggests that the simulated cloud-surface coupling is realistic. A joint-pdf analysis of relevant state
variables is conducted, suggesting that locations where the mixed-phase cloud layer is strongly coupled to the surface by convective updrafts act as hot-spots for invigorated interactions between turbulence, clouds and aerosol. A mixing-line analysis reveals that the turbulent mixing is similar to warm convective cloud regimes, but is accompanied by hydrometeor transitions that are unique for mixed-phase cloud systems. Distinct fingerprints in the joint-pdf diagrams also explain i) the typical ring-like shape of ice mass in the outflow cloud deck, ii) its slightly elevated buoyancy, and iii) an associated local minimum in
CCN.

## 1 Introduction

The ongoing accelerated warming of the Arctic climate is the result of various factors and feedback mechanisms, many of which are still poorly understood. The significance of the albedo feedback has long been known (Winton, 2006; Perovich et al., 2007), but recent research has highlighted the role of warm air intrusions (Bennartz et al., 2013) and the lapse rate feedback
(Pithan and Mauritsen, 2014; Lauer et al., 2020). Clouds play a sophisticated role in all of these mechanisms. For example, the albedo feedback can be mitigated by the persistent presence of reflective clouds in areas of retreating sea ice (Kay and



Gettelman, 2009; Hudson, 2011). Also, cloud presence in warm air intrusions significantly affects the downward long wave radiative flux at the surface (Liu et al., 2018). Even the lapse rate feedback is affected by clouds, with radiative cooling in mixed-phase cloud layers driving turbulent entrainment that counteracts larger scale subsidence, together acting to form and
maintain low-level thermal inversions (e.g. Neggers et al., 2019).

Arctic clouds form through a variety of processes, often taking place at small scales and involving many complex interactions between hydrometeors, aerosol and turbulence that are not yet fully understood (e.g. Mauritsen et al., 2011; Morrison et al., 2012). Gaining further insight has motivated intense research, including a number of field campaigns at high latitudes that were at least partially dedicated to this topic (Perovich et al., 1999; Tjernström et al., 2012; Knudsen et al., 2018; Wendisch
et al., 2019; Shupe et al., 2021). A large variety of cloud forms occurs at high latitudes; one possible way of classifying these can be based on the properties of the underlying surface. This yields three basic geographic regimes; i) clouds over relatively homogeneous sea ice, ii) clouds over fractional sea ice in the marginal ice zones, and iii) marine clouds over open water off the edge of the sea ice.

This study focuses exclusively on the third cloud category. More specifically, we focus on mixed-phase clouds in relatively
stagnant air masses over open water that are well coupled to the surface through convective transport. This choice is motivated by various factors. The ongoing shift in Arctic climate is perhaps strongest felt in areas where the sea ice disappears (Liu et al., 2012; Overland et al., 2014). The marine areas adjacent to the sea ice also act as gateways for injections of aerosol (Browse et al., 2014; Ito and Kawamiya, 2010), moisture and heat (Vázquez et al., 2016; Rinke et al., 2017; Pithan et al., 2018) into the high Arctic. Over open water the strong surface-atmosphere temperature difference can drive intense cloudy
convection, which is efficient in vertically mixing the lower atmosphere. While this phenomenon is well known from studies of cold air outbreaks (Atkinson and Wu Zhang, 1996; Fletcher et al., 2016; Gryschka and Raasch, 2005; Chlond, 1992; Müller et al., 1999), convection in more stagnant air masses has been investigated less thoroughly. However, such air masses are very frequent, perhaps even more so in a warmer future Arctic and its slower Polar jetstream (Screen et al., 2013; Barnes and Screen, 2015). Such slow-moving marine air masses have much more time to adjust to local conditions compared to fast-moving air
masses, making convection even more efficient in its vertical mixing.

Recent field campaigns have yielded extensive observational datasets on convection in stagnant marine Arctic air masses. To add value to such datasets and fill existing data gaps, an often-used method is to combine these with Large-Eddy Simulations (LES). LES stands for the numerical simulation of the discretized set of primitive equations for geophysical flow, at such high spatial and temporal resolutions that turbulence and convection are for the largest part resolved (Deardorff, 1970; Sommeria,
1976). Subgrid cloud physics can be included, possibly including multiple hydrometeors and phases of water, enabling the representation of mixed-phase clouds. Recent years have seen an increased use of LES experiments based on Arctic field campaign data, supplementing the observational data record and acting as a virtual research laboratory (e.g. Ovchinnikov et al., 2014; Stevens et al., 2018). At the same time, independent measurements of mixed-phase cloud properties can be used to evaluate the simulations (Kretzschmar et al., 2020; Ruiz-Donoso et al., 2020). Although LES does has its shortcomings, this
approach has led to demonstrable progress in understanding Arctic clouds. In particular process interactions that are hard to directly observe can well be studied using LES.





In this study, a composite LES case is constructed based on data collected by the Polar 5 and Polar 6 aircraft of the German Alfred Wegener Institute in the Fram Strait during the ACLOUD campaign (Arctic CLoud Observations Using airborne measurements during polar Day) (Wendisch et al., 2019). Research Flight 20 on 18 June 2017 sampled mixed-phase clouds

as embedded in a stagnant air mass off the sea ice edge, featuring significant convection over open water (but relatively weak compared to typical cold-air outbreak conditions). The first main objective of this study is to thoroughly evaluate the simulation against available measurements, covering thermodynamics, clouds and turbulence. The second main objective is to then use the simulation to gain insight into the role of cloud-surface coupling and aerosol-cloud-turbulence interactions in maintaining mixed-phase cloud layers during conditions typical for late spring/ early summer. The boundary conditions and large-scale

forcings are based on weather model analyses, while the initial conditions are based on in-situ dropsonde data. This yields a simulation that freely equilibrates towards the observed thermodynamic state. Key characteristics of the simulated mixed-phase clouds and convective transport in the boundary layer are evaluated against available aircraft measurements. Joint-PDF (Probability Density Functions) analyses in conserved variable space are then performed to identify and trace unique fingerprints of aerosol-cloud-turbulence interactions and hydrometeor transitions in simulated convective cells.

Section 2 describes details of the ACLOUD field campaign, including the weather situation, the flights of research aircrafts, and the observational datasets that were collected. The model configuration adopted in this study is described in detail in Section 3, including the LES code, the treatment of microphysics, the case configuration including forcings and boundary conditions, and the experimental setup. The presentation of the results is subdivided into two parts. Part I describes the basic behavior of the control experiment, including an evaluation of cloud and turbulence statistics against ACLOUD observational

datasets (Section 4). Part II then focuses on the joint-pdf analysis (Section 5). The obtained results are further interpreted in Section 6. Finally, in Section 7 the main conclusions are briefly summarized and an outlook is provided of future research inspired by this study.

## 2    Observations

### 2.1    The ACLOUD field campaign

The ACLOUD field campaign took place from 23 May to 26 June 2017 in the vicinity of Svalbard. ACLOUD and its sister campaign PASCAL (Physical feedbacks of Arctic planetary boundary level Sea ice, Cloud and AerosoL; Macke and Flores, 2018; Wendisch et al., 2019) were part of the ongoing $(\mathcal{AC})^3$ research program (Arctic Amplification: Climate Relevant Atmospheric and Surface Processes and Feedback Mechanisms; Wendisch, 2017). Both campaigns focused on clouds in the lower troposphere in the northern Fram Strait area in Arctic Spring. ACLOUD featured collocated airborne observations

(Ehrlich et al., 2019b) performed by the aircraft Polar 5 (P5) and Polar 6 (P6) of the German Alfred-Wegener-Institut (Wesche et al., 2016).

During the ACLOUD campaign, three distinct synoptic weather regimes can be distinguished (Knudsen et al., 2018). The first week was characterised by a northerly inflow of cold and dry air. This resulted in prevailing low-level clouds over open water. During the following two weeks, the prevailing air masses originated from the South and the East. The lower troposphere



**Figure 1.** MODIS TERRA true-color image at 250m effective resolution of the Fram Strait west of Svalbard during RF20 on 18 June 2017. The Svalbard landmass is visible on the right, while the sea-ice and its margin are situated in the north. The flight tracks of Polar 5 and 6 are shown in red and yellow, respectively. For P5 the waypoints (C), Polarstern (PS) and Longyearbyen airport (LYR) are also indicated, with the dropsonde locations (DS) shown as blue dots. The 24-hour air mass trajectory intersecting with DS04 is shown as a sold green line. The target domain including the southern race track section is indicated by the dotted orange box, of which the simulated part is indicated by the dotted green box. MODIS data obtained through NASA WorldView (https://worldview.earthdata.nasa.gov/).





was significantly warmer and moister, with variations in the cloud cover. The final two weeks were then dominated by a
westerly flow, featuring considerable variations in cloud cover and temperature in the mid-troposphere. Airborne observations
were made during a wide range of cloud conditions, including both stably stratified and convective regimes. Both single-layer
and multi-layer clouds structures were observed, but also clear-sky conditions.

## 2.2 RF20

This study exclusively focuses on Research Flight **RF20** by the P5 and P6 aircraft on 18 June 2017. This flight took place in
the Fram Strait west of Svalbard (see Figure 1). As described by Knudsen et al. (2018), during this period the mid-troposphere
experienced drying. High cirrus clouds were present during the previous day but disappeared on 18 June. The air mass in the
Fram Strait was slow-moving on this day, being situated over relatively warm open water.

Figure 1 illustrates that the cloud situation in the Fram Strait as encountered by the aircraft was relatively complex. However,
a closer look suggests that a rough north-south regional division in cloud character can be made. In the north, over the sea ice
and its margin, the clouds were absent or very thin, allowing good visibility of the sea ice from the satellite and the aircraft.
In the western and middle parts of the Fram Strait the clouds were thicker but still only weakly convective, visible in Fig. 1 as
vague but not completely opaque cloud patches. In the southern part the clouds were truly convective, being broken, thicker
and more opaque.

The P5 and P6 aircraft followed anti-clockwise flight paths from their base in Longyearbyen (LYR) and visited these three
regimes consecutively. This study focuses exclusively on the convective clouds in the southern areas, as sampled during the
last eastbound flight leg between waypoints C5 and C6. In this section, also referred to as the "southern race-track", both
aircraft flew back and forth between C5 and C6. While the P5 doubled back once and maintained altitude above the boundary
layer inversion (at about 1.5 km height), the P6 doubled back twice, staying below inversion height. It maintained constant
altitude for five brief segments to allow covariance calculations at multiple heights, as shown in Appendix A. In-situ in-cloud
measurements were also made by the P6 during this period (see Fig. A1). The enhanced and targeted sampling during the
southern race-track sections, as well as the occurrence of significant mixed-phase convection, motivates adopting this area as
the target domain of this study, as indicated by the orange box in Fig. 1.

## 2.3 Observational datasets

The observational data from the ACLOUD campaign used in this study are summarized in Table 1. Figure 2a shows detailed
measurements of the clouds in the target area as obtained with the Microwave Radar and radiometer for Arctic Clouds radar
onboard P5 (MiRAC; Mech et al., 2019). Because of the doubling-back between C5 and C6, the same cloud structure appears
three times, once in each panel. Cloud top height varies significantly along the flight track in this area, which is typical for
broken convective cloud fields. The maximum cloud top height is at approximately 1.4 km, which is consistent with the thermal
inversion height visible in the DS04 profile (see Fig. 3a) and the Airborne Mobile Aerosol Lidar (AMALi; Stachlewska et al.,
2010) measurements (also included in Fig. 2). The MiRAC flight sections between 7-8.5 °E feature significant but narrow
convective precipitation that also reached the surface. This area was visited by the P5 three times, at around 16:20, 16:35 and



**Figure 2.** Time-height cross section of P5 MiRAC radar reflectivity $Z_e$ during RF20 (contour shading) and the indicated liquid cloud top from AMALi (black dots). The displayed longitude range corresponds to the orange target domain as shown in Fig. 1, while the lime green horizontal line indicates the simulated domain. The black arrows indicate the flight direction of the aircraft for each leg, with the start and end times indicated at the sides. The location of the dropsonde DS04 is indicated by the dotted dark green vertical line.





**Table 1.** Overview of the observational datasets used in this study. Data in rows 1-7 are accessible through the PANGAEA database, while the MODIS data is available through the NASA WorldView[a] interface.

| Instrument | Platform | Description | Variables | Reference |
|---|---|---|---|---|
| Dropsondes | P5 | RS904 | Thermodynamic state | Ehrlich et al. (2019a) |
| MiRAC | P5 | 94 GHz cloud radar and 89 GHz radiometer | Cloud vertical structure | Kliesch and Mech (2019) |
| AMALi | P5 | Lidar at 532 nm | Cloud boundaries | Neuber et al. (2019) |
| UHSAS | P6 | Aerosol mass spectrometer | Aerosol concentrations | Mertes et al. (2019) |
| UHSAS-2 | P6 | Aerosol mass spectrometer | Aerosol concentrations | Zanatta and Herber (2019) |
| Nevzorov probe | P6 | Hot-wire probe | Liquid water content | Chechin (2019) |
| Cloud Imaging Probe | P6 | Optical Array Probe | Ice water content (CIP_IWC_BF95) | Dupuy et al. (2019) |
| Noseboom | P6 | Eddy covariance (100 Hz) | Turbulent heat flux | Hartmann et al. (2019) |
| NASA MODIS | Terra, Aqua | Spectroradiometer | 2-band reflectance (250m res) | Savtchenko et al. (2004) |

[a] https://worldview.earthdata.nasa.gov

17:00 UTC. The DS04 dropsonde was also launched into this area. Freezing level is situated at about 350 m height, which is well below the maximum cloud top.

The eastern part of the target domain that is centered around dropsonde DS04 is selected as the area to be simulated (indicated by the green box in Fig. 1 and the green line in Fig. 2). This choice is motivated by the following considerations. First, our main objective is to study interactions between convection, mixed-phase clouds and aerosols, which did occur in this area. Second, the combination of warm water and cold air implies large surface latent and sensible heat fluxes, making the near-surface convection vigorous and potentially well coupled to the cloud layer. Such convective conditions often occur in this

region, for a large part controlled by the wind direction. Third, convective clouds are well resolved in Large-Eddy Simulations (LES). A further advantage is that the cloud-bearing low-level air mass in this area was slow-moving and almost stagnant. This is evident from i) the dropsonde profiles of $u$ and $v$ (Fig. 3) and ii) the trajectory staying in the proximity to the dropsonde location for about 24 hours (Fig. 1). This broadens the time span that the simulation results can be justifiably compared to relevant measurements. The P5 and P6 aircraft visited the area between waypoints C5 and C6 multiple times, which enhances

the sample size. Finally, the decision to only simulate the eastern part of the target domain is mainly based on the need to avoid averaging large-scale forcings over a too wide area, so that the simulated convection remains optimally representative of the local (convective) conditions surrounding DS04.

The DS04 dropsonde data provide detailed insight into the thermodynamic structure of the marine boundary layer in the simulation domain. Figures 3a and b show that an inversion layer is present between $1.3 - 1.5$ km height, and can be recog-

nized in most state variables. A well-mixed sub-cloud layer of about 350 m depth is capped by a relatively deep cloud layer, characterized by high relative humidity values and a conditionally unstable thermodynamic vertical structure. The inversion features a $\theta_v$-jump of about $+3$ K. In contrast, the jump in water vapor specific humidity $q_v$ is relatively small, indicating the





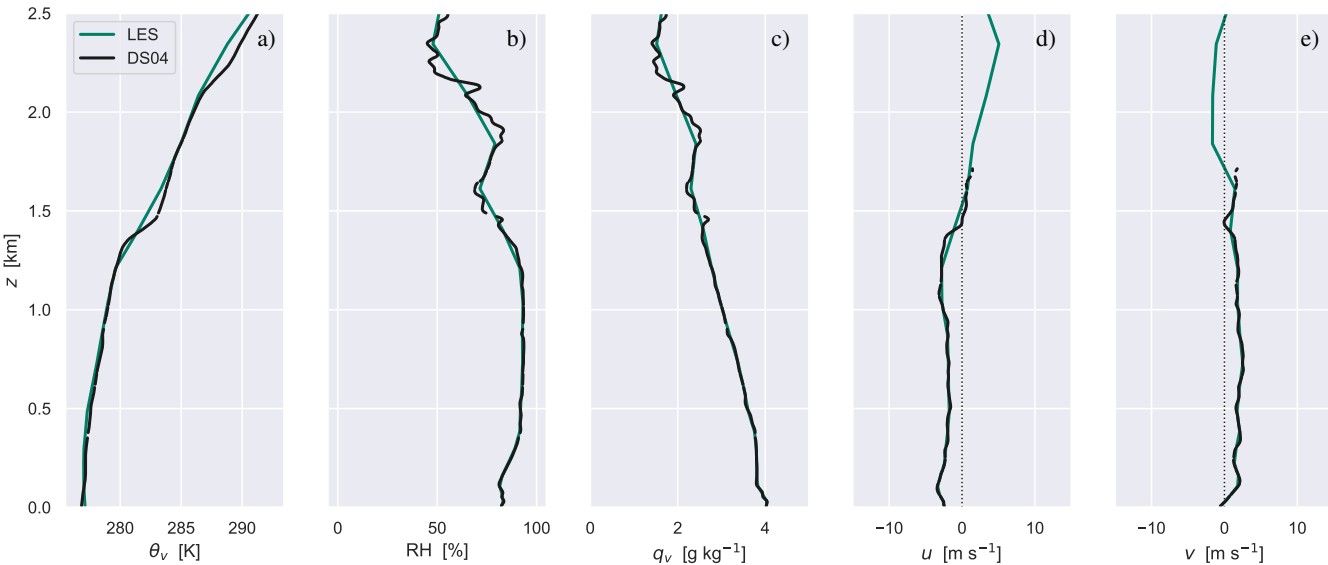

**Figure 3.** Dropsonde profiles from RF20 of a) virtual potential temperature $\theta_v$, b) relative humidity RH, c) water vapor specific humidity $q_v$, and the d) zonal and e) meridional wind speeds $u$ and $v$, respectively. The black line indicates the DS04 sounding, while the green lines indicate the idealized initial profiles of the LES experiment.

presence of significant water vapor above the inversion. There is a notable gap in wind measurements between $1.6 - 2.5$ km, where samples have been removed after quality checks.

Cloud measurements are of key importance for this study, given its focus on understanding interactions between microphysics, turbulence and aerosol in mixed-phase convective clouds. In particular, the observational data on hydrometeor occurrence and mass as collected by the P5 and P6 are useful for evaluating LES experiments. While the MiRAC and AMALi data onboard P5 provide information about cloud top heights, vertical structure, and liquid water path, the Nevzorov probe onboard P6 provides in-situ samples of cloud liquid water content. The LaMP CIP probe provides information on ice water content us-

ing Brown and Francis (1995) mass diameter relationship on non-spherical particles only. Finally, the high-frequency (100 Hz) turbulence measurements collected by the P6 noseboom (Hartmann et al., 2018) allow calculating (co)variances of temperature and vertical velocity in the boundary layer, even for relatively short flight segments.

During RF20 the P6 carried two Ultra-High Sensitivity Aerosol Spectrometers (UHSAS) to sample submicron-size particles (Zanatta and Herber, 2019). These instruments measured the number size distribution of particles with diameters between

60 nm and 1000 nm by detecting scattered laser light, using the method described in detail by Cai et al. (2008). Figure 4 shows that a relatively wide range of aerosol concentrations was encountered. The data suggests a marked difference in aerosol loading below and above the cloud layer, with the lower values found below the clouds. These measurements as documented





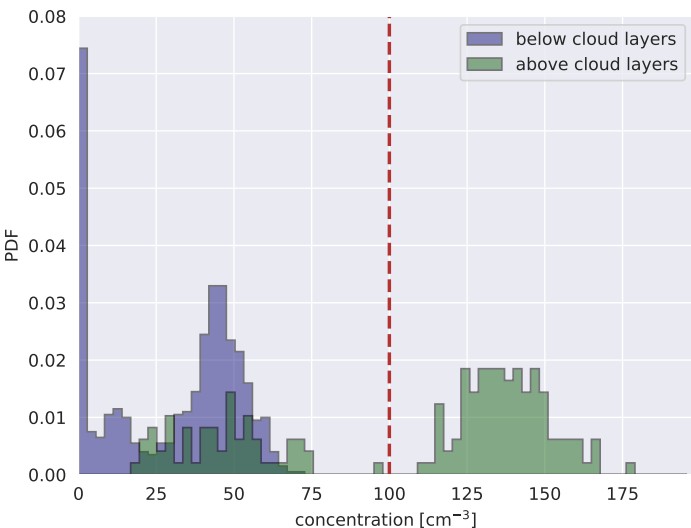

**Figure 4.** The histogram of aerosol concentrations (for aerosol particles of diameters between 80 nm and 1000 nm) as observed by the USHAS-2 mass spectrometer on P6 during the southern race track section of RF20. The red dashed line indicate the initial CCN concentration used in the simulations. (Zanatta and Herber, 2019; Mertes et al., 2019).

by Mertes et al. (2019) form the basis for the initial CCN profile as adopted in the simulations described in Section 3.4. The skill of the model in reproducing the observed vertical structure of aerosol will be assessed.

## 3   Model configuration

### 3.1   DALES

The simulations in this study are carried out with the Dutch Atmospheric Large Eddy Simulation model ((DALES, Heus et al., 2010). DALES has been successfully applied to simulate observed turbulent/convective boundary-layers and clouds in many climate regimes, including the tropics (Vilà-Guerau de Arellano et al., 2020), the subtropics (Van der Dussen et al., 2013; de Roode et al., 2016; Reilly et al., 2020), mid-latitudes (Neggers et al., 2012; Corbetta et al., 2015; Van Laar et al., 2019) and high latitudes (de Roode et al., 2019; Neggers et al., 2019; Egerer et al., 2020; Neggers, 2020a, b). The code of DALES is open source and maintained online at *https://github.com/dalesteam/dales*. The governing equations, numerical aspects and the various subgrid physics packages are described in detail by Heus et al. (2010), and accordingly only a brief summary is provided here. At the foundation of the model are the Ogura-Phillips anelastic equations for a set of prognostic variables including the three velocity components $\{u, v, w\}$, total water specific humidity $q_t$, liquid water potential temperature $\theta_l$, as well as the number and mass of various hydrometeor species. The time-integration makes use of a 3rd-order Runge-Kutta approach.





Scalar advection is represented using an upwind scheme, while the centered difference method is applied for the three velocity components. The subgrid-scale transport of heat, moisture and momentum is dependent on a prognostic TKE model. For radiation a multi-waveband transfer model is used in combination with a Monte Carlo approach (Pincus and Stevens, 2009). A

new mixed-phase cloud microphysics scheme was recently implemented, as described in more detail in the next subsection.

## 3.2   Microphysics

The control version of DALES (Heus et al., 2010) includes a double moment microphysics scheme for warm clouds featuring two hydrometeors, cloud water and rain (Seifert and Beheng, 2001). To simulate Arctic mixed phase clouds the mixed-phase extension described by Seifert and Beheng (2006) (hereafter SB06) was recently implemented, adding a further three prog-

nostic hydrometeors (cloud ice, snow and graupel). First DALES results with this scheme, including an evaluation against Polarstern cloud measurements during the PASCAL campaign in 2017, are described by Neggers et al. (2019). In principle the implementation in DALES closely follows SB06. In this section some details of the implementation are described that are either i) different from the SB06 description or ii) are particularly relevant for this study.

A key difference with SB06 is the prognostic treatment of the number concentration of Cloud Condensation Nucleii (CCN).

This first applies to activation of CCN in saturated grid cells. The CCN concentration is conserved during nucleation of cloud droplets, their condensational growth and evaporation. The sedimentation of cloud droplets contributes together with the convection to the vertical transport of CCN. The self-collection of cloud droplets and precipitating processes act as a sinks for CCN. For simplicity the collection of cloud droplets by ice particles and the freezing of cloud droplets are also treated as CCN sink terms. The glaciation of clouds does not cause CCN depletion, because in the Wegener-Bergeron-Findeisen regime the

vapor deposition on growing ice crystals evaporates liquid water but leaves the surrounding CCN unaffected (Schwarzenböck et al., 2001).

The primary ice production in SB06 accounts for ice nucleation, as well as freezing of cloud droplets and raindrops. Ice nucleation follows the parameterization proposed by Reisner et al. (1998), prescribing a constant number concentration of the available Ice Nucleating Particles (INP) and ignoring any removal (see Appendix C). The freezing of liquid hydrometeors is

described by the stochastic model proposed by Bigg (1953). The secondary ice production accounts for ice multiplication by the Hallet-Mossop process (Hallett and Mossop, 1974), occurring during the riming of ice hydrometeors in the temperature ranges between 265 K and 270 K (Griggs and Choularton, 1986; Beheng, 1982). Other mechanisms of secondary ice production are not considered. Processes modifying the number and mass of ice hydrometeors include deposition, riming, aggregation of snow, self-collection of snow, partial conversion of snow and ice crystals to graupel, collection of snow by graupel, sublimation,

melting, evaporation, and enhanced melting (i.e. melting due to collisions with liquid hydrometeors in temperatures above freezing point). The contributions to number and mass tendencies by these microphysical processes are calculated in the order established by Seifert and Beheng (2006b).

The majority of parameters in the DALES microphysics scheme follow the control setup of SB06, with the exception of the values of coefficients for shape and velocity of cloud ice. These were adjusted to the same values as adopted in the recent

intercomparison study on a marine cold air outbreak by de Roode et al. (2019) to better reflect conditions in Arctic low-level





clouds. This decision is also motivated by the fact that both cases describe conditions in more or less the same region. The full setting of microphysical parameters adopted in this study is provided in Appendix C and Table C1.

### 3.3 Initialization, boundary conditions and composite forcing

The back trajectory calculated from the time and location of the DS04 dropsonde indicates that the air mass did not move
within a degree of this location in the time period between 00 UTC and the dropsonde launch (see Fig. 1). This reflects the approximately stagnant wind conditions as also detected by the DS04 dropsonde (see Fig. 3). In addition, the large-scale conditions did not change much on this day. These conditions motivate adopting a time-composite case setup that reflects large-scale conditions in the region as averaged over the twelve-hour period leading up the DS04 launch.

Large-scale data from the Integrated Forecasting System (IFS) of the European Centre for Medium-range Weather forecasts
(ECMWF) are used to represent the impacts of larger-scale phenomena during the simulation. Following Van Laar et al. (2019), a combination of analyses (available every 12 hours) and short-range forecasts (available every 3-hours) is used, effectively yielding a four-dimensional dataset of the atmospheric state variables $\{\theta_l, q_t, u, v\}$ at 3-hourly temporal resolution and 0.1x0.1 degree spatial resolution. In this study these are calculated at 3-hourly points along the back-trajectory, a method previously adopted by Neggers et al. (2019). The forcing profiles are time and height dependent. Horizontal advective forcings
are represented as prescribed advective tendencies, calculated through horizontal averaging within a $0.5° \times 0.5°$-wide column around the location. The tendency due to large-scale subsidence relies on a prescribed profile of pressure velocity that acts on the evolving vertical structure in the LES. Forcings in the momentum equation include the Coriolis term and the pressure gradient term, in combination expressed as the departure of the model wind from the prescribed geostrophic wind. The latter is calculated from the pressure field. Given these time- and height-dependent forcing profiles at the trajectory points, time-
averaging is then applied over 12 hours to obtain the composite forcing dataset used to drive the LES. These profiles are described in more detail in Appendix B), and are characterized by a persistent low-level subsidence, a weak advective cooling and moistening tendency above 1 km height, and negligible geostrophic forcing of the wind.

The DS04 dropsonde profiles are used as initial state, amalgamated with the composite large-scale model data. Where available, the sonde data is averaged onto the vertical grid of the composite ECMWF forcings. At grid layers where no DS04 data
is available, the composite model state itself is used instead. Figure 3 shows the initial profiles thus obtained, illustrating that the method successfully yields profiles that are continuous and do not include huge jumps that could result from mismatches between sonde and ECMWF data.

The surface boundary conditions include a prescribed skin temperature and humidity. The latter is calculated by assuming oceanic ice-free conditions, so that the associated saturation specific humidity can be used. These skin values are then used
to interactively calculate the surface fluxes of heat, moisture and momentum, using prescribed roughness lengths for heat, moisture and momentum. The calculation of the bulk drag and exchange coefficients relies on the stability functions that are native to the DALES code (Heus et al., 2010). A prescribed surface albedo is used to calculate the upward short wave radiative flux at the lower boundary. The incoming short wave radiative flux at the model ceiling depends on i) seasonality and time of



**Table 2.** Summary of defining characteristics of the control LES experiment for RF20.

|  | Description | Unit | Value |
|---|---|---|---|
| $\Delta x, \Delta y, \Delta z$ | Grid spacing | m | $50 \times 50 \times 40$ |
| $\Delta t$ | Time step | s | Adaptive |
| $L_x, L_y, L_z$ | Domain size | km | $25.6 \times 25.6 \times 5$ |
| $\alpha$ | Surface albedo | - | 0.06 |
| $T_{\mathrm{skin}}$ | Skin temperature | K | 278.68 |
| $z_0^{\mathrm{m}}$ | Roughness length (momentum) | m | 1.6e-4 |
| $z_0^{\theta}, z_0^{\mathrm{q}}$ | Roughness length (heat and moisture) | m | 3.2e-5 |
| $p_{\mathrm{s}}$ | Surface pressure | hPa | 1007.54 |
| CCN | Initial CCN number concentration | $\mathrm{cm}^{-3}$ | 100 |
| INP | Upper limit for INP number concentration | $\mathrm{m}^{-3}$ | 1000 |

day and ii) the composite large-scale state above the model ceiling. In doing so we follow the method adopted by Van Laar
et al. (2019).

All large-scale forcings are time-constant and are applied in a horizontally homogeneous way, being the same in every LES
grid column. In addition, horizontally periodic boundary conditions are applied. The resulting simulation can thus be interpreted
as a statistical down-scaling of the dropsonde profile, with the LES acting as a generator of the small-scale variability existing
in the domain around it.

## 3.4  Control experiment

This study makes use of one control simulation, designed to match the observed thermodynamic and cloudy state as closely
as possible. This experiment is to serve as a benchmark simulation for planned subsequent studies (not covered in this paper).
Table 2 summarizes the main characteristics of this experiment. The size of the simulated domain is considered wide and high
enough to accommodate the typical width of convective structures observed in the Arctic (Müller et al., 1999). The duration
of the experiment is 72 hours, to provide the turbulent boundary layer with enough time to equilibrate and to cover three full
diurnal cycles. Radiation is interactive with all five hydrometeors. Above the turbulent domain the composite ECMWF profile
is used in the calculation of the downward fluxes at the model ceiling. The solar inclination is time- and latitude dependent,
introducing a diurnal cycle in the radiation.

A sponge layer is applied in the top 1 km of the computational domain to prevent reflection of waves off the rigid top
boundary. In addition, continuous nudging towards the initial profile is applied above the boundary layer inversion to prevent
excessive model drift in this height range. A Newtonian relaxation term is included in the prognostic equations for $\{u, v, \theta_l, q_t\}$
to this purpose, adopting a timescale of 6 hours. A 200 m deep transition zone is included above the inversion across which the
nudging intensity increases linearly with height. Note that in this configuration the resolved turbulence and convection below
the inversion remains unaffected by the free-tropospheric nudging, and can freely equilibrate in response to the initial- and





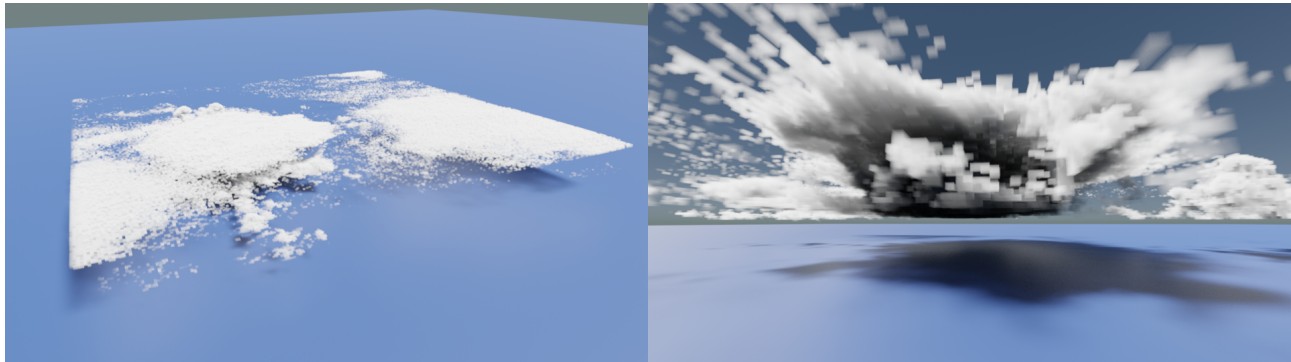

**Figure 5.** Three-dimensional volume renderings of cloud liquid water $q_l$ in a snapshot from the control simulation with DALES for ACLOUD RF20. The Blender tool is used to perform the ray tracing, including two shaders representing scattering and absorption, both dependent on cloud mass in each gridcell ($50 \times 50 \times 40$ m$^3$). Left: birds-eye perspective of the simulated domain ($25.6 \times 25.6$ km$^2$). Right: close-up view of a simulated convective cell.

boundary conditions and the prescribed time-constant forcings. Such nudging has successfully been applied in previous LES studies in which equilibration played an important role (Sandu et al., 2009), motivating adopting this concept here.

The observed statistical distribution of aerosol concentration as shown in Fig. 4 informs the initial CCN profile adopted in the control experiment, chosen here to be constant with height at $100$ cm$^{-3}$. This choice is motivated by the following arguments. Firstly, it is safe to assume that only a fraction of the observed aerosol can act as CCN, here assumed to be approximately

50-90 %. Secondly, CCN is treated prognostically in the model and can evolve freely during the simulations, and no external sources of CCN are considered for simplicity. With the convection and clouds gradually removing aerosol below the inversion, a choice of $100$ cm$^{-3}$ should after some time result in a vertical structure resembling the observed one as shown in 4, with concentrations below the clouds being about half of the values above. This will be assessed in the model evaluation.

The microphysics scheme is operated using a maximum initial cloud droplet number concentration (CDCN) in supersat-

urated areas of $50$ cm$^{-3}$. The number concentration of INP is assumed time-constant and prescribed at $1000$ m$^{-3}$, therein following Reisner et al. (1998) and Hartmann et al. (2019).

## 4   Results I: Evaluation

### 4.1   Liquid cloud macrophysics

Figure 5 shows a volume rendering of a three-dimensional snapshot of cloud liquid water during the control experiment with

DALES of the composite RF20 case. Two cloud macrophysical characteristics stand out that define this case. The first is the presence of clusters of opaque liquid cloud objects that have a flat cloud base (right panel). These are surface-driven convective cloud structures, which were also observed during the flight. The second key feature is the presence of wide and

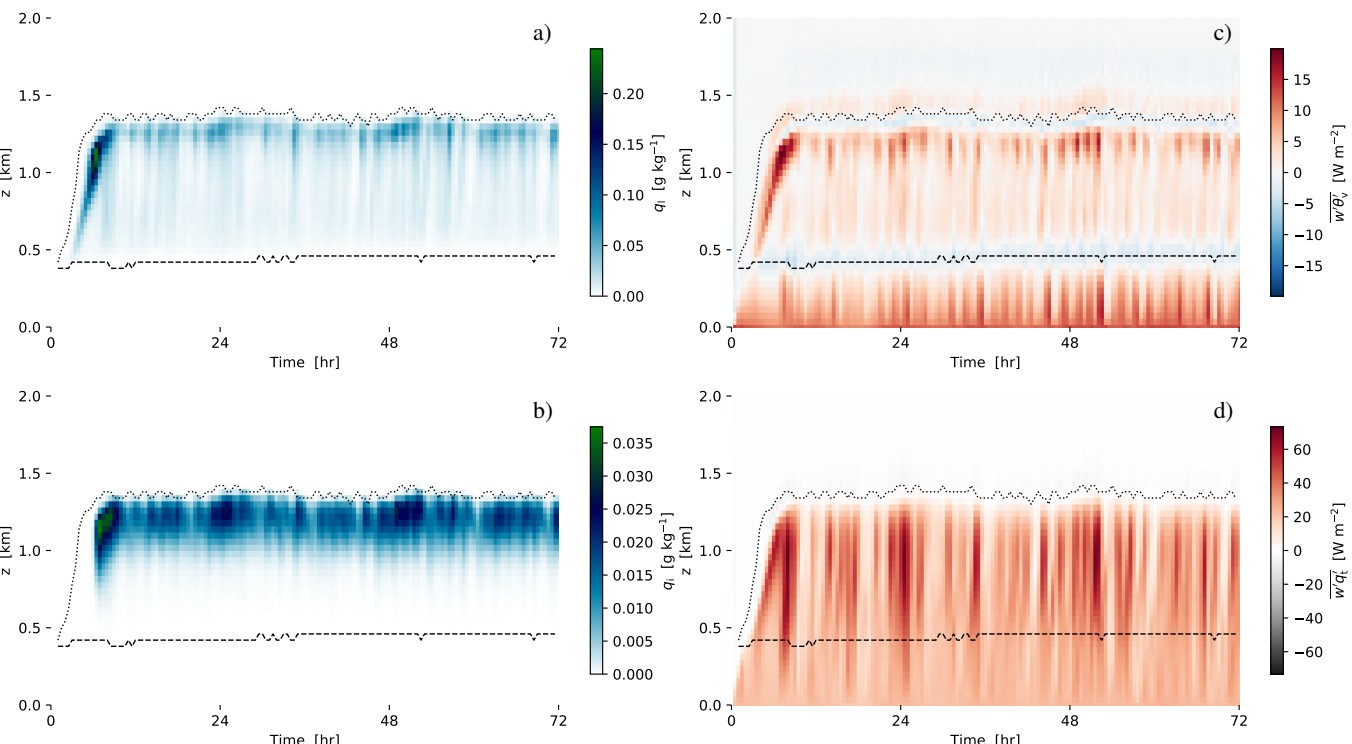

**Figure 6.** Time-height contour plots of domain-averages of four selected variables. a) Cloud liquid water $q_l$, b) cloud ice water $q_i$, c) the turbulent $\theta_v$ flux and d) turbulent $q_t$ flux. Averaging time is 30 min. The dashed and dotted lines always reflect the lowest base and highest top of liquid clouds in the domain, respectively.

thin outflow layers in liquid phase that are attached to the largest convective objects (left panel). This anvil-shaped structure of the liquid cloud field (see also Fig. 12) is a defining feature of surface-driven warm convective boundary-layer clouds under

strong inversions (Stevens et al., 2001; Neggers et al., 2017; Stevens et al., 2020). It is also distinctly different from turbulent mixed-phase clouds over homogeneous sea ice, which are often fully decoupled (e.g. Solomon et al., 2014). What sets these convective boundary layer cloud systems apart from their equivalents in warmer climes is the presence of cloud ice (not shown in this rendering).

## 4.2 Time evolution

Figure 6a-b document the time development of the domain-averaged cloud structure and phase during the control simulation of the composite RF20 case. After a spin-up period of about 12 hours the boundary layer more or less equilibrates, staying close to this state for the remainder of the simulation. The cloud layer is in mixed-phase, with liquid cloud coexisting with ice cloud. Both have a maximum immediately below the inversion. A weak diurnal cycle in cloud mass, cloud thickness and inversion height can be distinguished that is superimposed on the long-term equilibrium. This diurnal behavior is well known





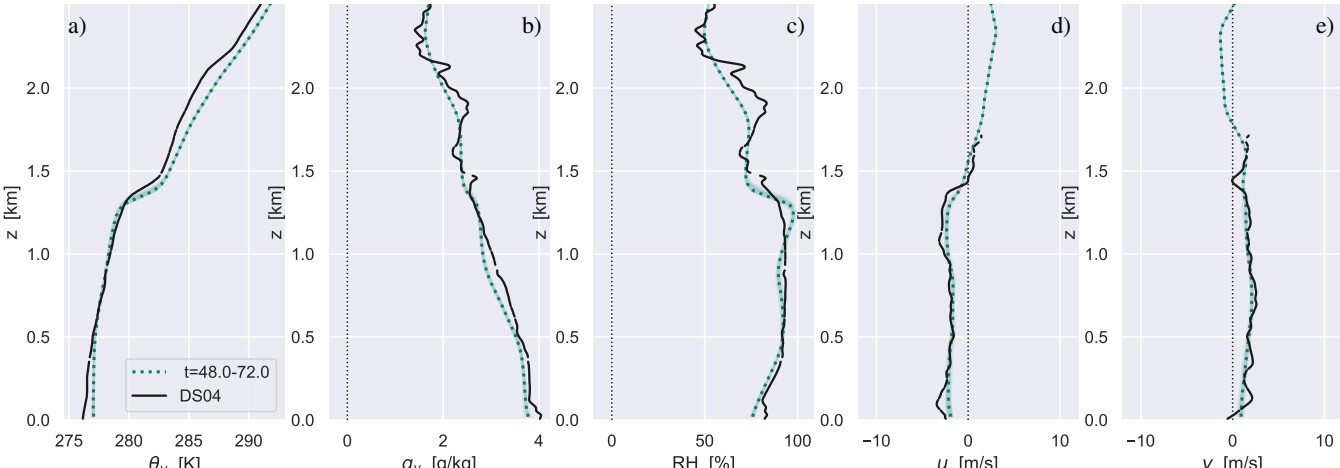

**Figure 7.** Domain- and time-averaged vertical profiles of a) virtual potential temperature $\theta_\mathrm{v}$, b) water vapor specific humidity $q_\mathrm{v}$, c) RH, d) zonal wind speed $u$ and e) meridional wind speed $v$. The DS04 dropsonde sounding is shown in black. The time averaging covers the final full-day period of the simulation (48-72 hr). Shading in the profile plots represents the spread among half-hourly domain-averages, with contour lines representing the percentiles $\{5, 25, 50, 75, 95\}$ %.

from warm marine stratocumulus, and is driven by daytime absorption of short wave radiation alternating with nighttime cloud top cooling (Rozendaal et al., 1995; Wood et al., 2002; Duynkerke et al., 2004). The presence of this signal here indicates that in early June the solar radiation is already strong enough at these latitudes to drive a boundary layer response.

    Figure 6c-d show the total (resolved + subgrid) fluxes of virtual potential temperature and total specific humidity. Significant boundary layer-deep transport is present, reflecting a high degree of coupling between the cloud layer and the surface. The

evolution of the humidity flux shows that it takes about 12 hours for the surface-driven turbulence to properly spin up after initialization. Once this has occurred the transport is continuous, with the local minimum in the $\theta_\mathrm{v}$ flux near liquid cloud base indicating the presence of a shallow transition layer (see Fig. 6c). Such stable layers with CIN (Convective Inhibition) are a well-known feature of cumulus-capped boundary layers (Albrecht et al., 1979) and decoupled stratocumulus (Nicholls, 1984). The transport intensity is intermittent at times, which could reflect the impact of subsampling of convective events due to a still

limited domain size.

### 4.3 Vertical structure

The simulated vertical profiles of various state variables are compared to the dropsonde soundings in Fig. 7. Only those variables are shown for which measurements are available. In general the simulated profiles agree well with the observations concerning the vertical structure of the boundary layer, with a relatively well-mixed layer up to about $500$ m capped by a cloud

layer featuring conditionally unstable thermodynamic lapse rates. This cloud layer with high relative humidity extends up to





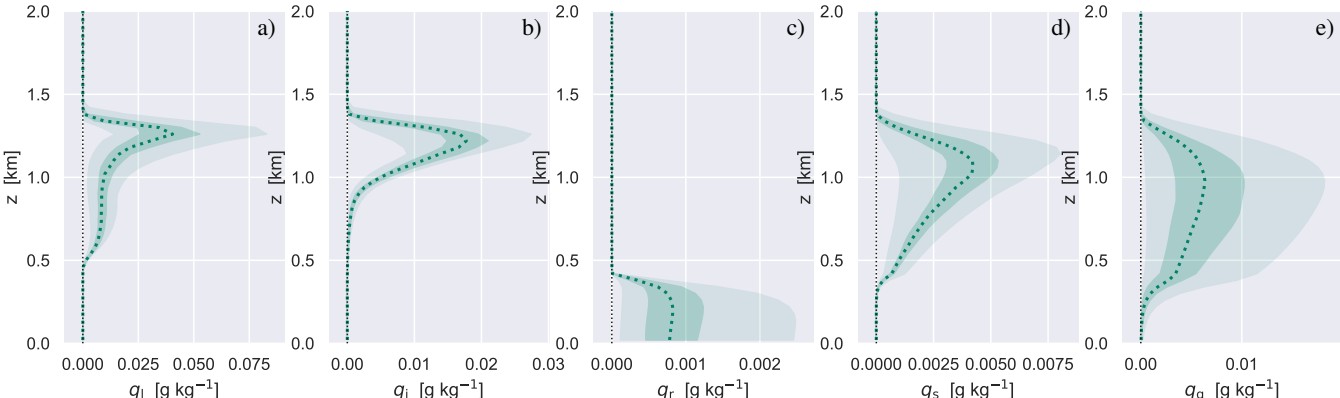

**Figure 8.** Same as Fig. 7 but now showing the specific masses of all five hydrometeor species in the LES, including a) liquid water $q_l$, b) ice water $q_i$, c) rain $q_r$, d) snow $q_s$ and e) graupel $q_g$.

about 1.4 km. The inversion layer of a few hundred meters deep is also reproduced well, featuring a $\theta_v$ jump of about +3 K and a negligible $q_v$ jump. Similar to the observations the wind amplitude is weak throughout the lower atmosphere.

Figure 8 shows vertical profiles of all five hydrometeor species included in the DALES microphysics scheme. Their structure gives insight into their origin and in the transition between suspended and falling species. Cloud liquid water $q_l$ has a distinct

mode near the inversion but still has significant values below, reflecting the updraft-anvil cloud structure visible in Fig. 5. Cloud ice $q_i$ peaks near the inversion at approximately the same height, and decreases below and disappearing a few 100 m above the liquid cloud base. Such amplitudes in liquid and frozen cloud condensate have been reported in previous studies of convective mixed-phase clouds over open water in the Arctic (Klein et al., 2009; Morrison et al., 2009). Some rain $q_r$ is present below liquid cloud base, but snow $q_s$ and graupel $q_g$ masses are much larger, peaking at a slightly lower height compared to

the suspended hydrometeors. This feature, in combination with the almost discrete change at $z \sim 400$ m from snow/graupel to rain, suggests that significant precipitation forms near the inversion and then forms shafts extending to the surface. Precipitation melts when it crosses freezing level, losing mass on its way down due to sublimation and evaporation. Such shafts are actually visible in the radar measurements shown in Fig. 2.

### 4.4 Cloud boundaries and hydrometeors

Figure 9 compares the simulated cloud top heights against those derived from the downward-pointing MiRAC and AMALi onboard P5. A statistical approach is adopted, making use of pdfs of cloud top height. LES cloud top height is established by searching for the first model level below the P5 cruising altitude at which hydrometeor mass exceeds a near zero threshold ($10\mathrm{e}^{-6}$ kg/kg). This is performed for all $(x,y)$ locations on the grid, yielding a pdf that should be conceptually comparable to the aircraft measurements. Liquid and ice clouds are considered separately, given the fact that their spatial structure is so

different.





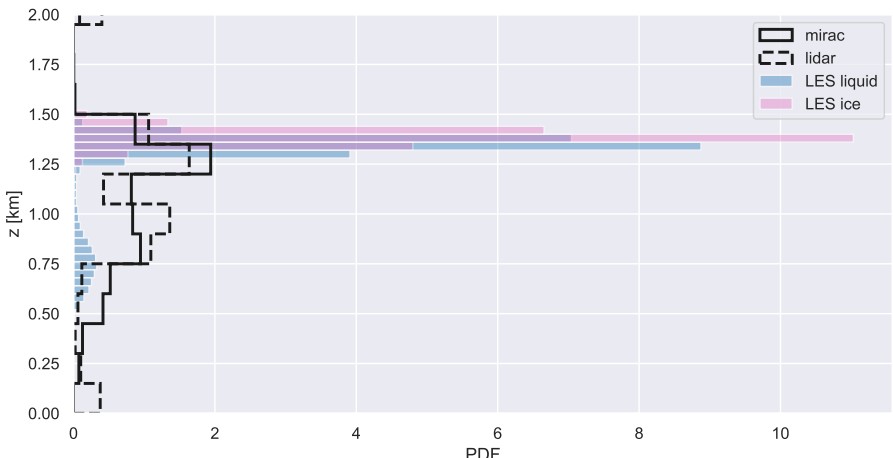

**Figure 9.** Vertical distributions of simulated (colored) and observed (black) cloud top heights. All samples are sorted into normalized histograms, shown as bar plots. The liquid and ice cloud top heights diagnosed in LES are shown in blue and pink, respectively. The bin width is equal to the vertical discretization of the LES experiment. The values detected by the P5 Mirac and Lidar during the southern race track are shown as solid and dashed black lines, respectively. These data are already shown in Fig. 2.

The results indicate that the maximum height at which hydrometeors are observed is well reproduced by the model. This is consistent with the good agreement on inversion height as detected by the dropsonde. The shape of the cloud top distribution is less well reproduced, although some liquid cloud tops are present in the model above $z \approx 500$ m. If this is a model shortcoming, or rather results from the small observational sample size due to the sampling of only one convective event, remains a question that can not be conclusively answered at this point. For example, the convective structure that was observed might have been rare, so that the observations are skewed towards the associated cloud properties.

Figure 10 confronts LES with in-cloud measurements by the Nevzorov probe (liquid hydrometeor mass) and the CIP probe (frozen hydrometeor mass). While the samples are shown as individual dots, the LES results are shown as a pdf, calculated from the values of all gridboxes on a horizontal slice. Adopting this evaluation method is motivated by the fact that the measurements are only available for a single flight track. Using pdfs as shown in Fig. 10 provides a bandwidth for interpreting the probability of such point samples, while still preserving information about extremes. The comparison is thus not quantitative, but should only be interpreted as indicating if the simulated range of values is realistic or not. In other words, when the measurements fall inside the pdf of LES values, then it is theoretically possible to encounter such values when randomly sampling the LES domain. This can be considered a measure for the representativeness of the simulation of the observed clouds.

In each panel the simulated and observed values include all hydrometeor species in the phase of interest. For the LES this means $q_l + q_r$ for liquid and $q_i + q_s + q_g$ for ice. This choice is motivated by the fact that the definition of these species in the microphysics scheme is hard to match with the measurements. By adding all species for each phase, such potential mismatches





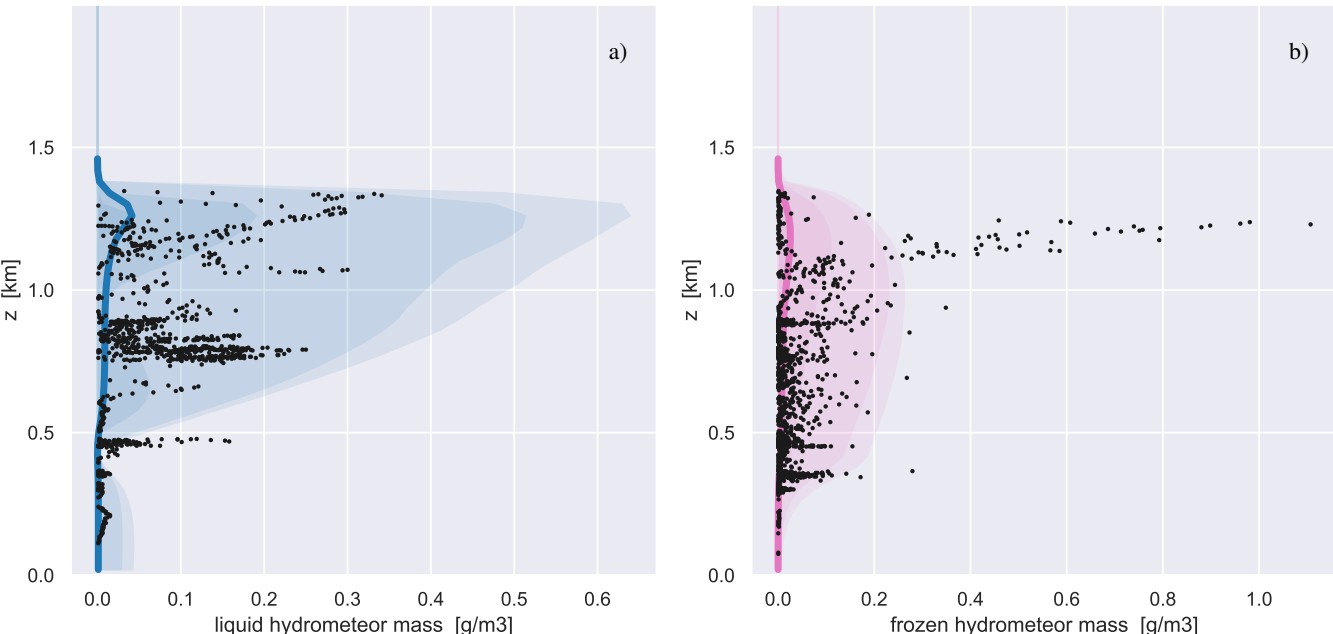

**Figure 10.** Simulated and observed hydrometeor mass in a) liquid phase and b) ice phase. Measurements by the P6 Nevzorov probe and Cloud Imaging Probe (CIP) are shown as black dots, only including samples during the southern race track segment of RF20. a) Sum of all liquid-phase hydrometeors, $q_l + q_r$. b) Sum of all ice-phase hydrometeors, $q_i + q_s + q_g$ in b). The distribution of LES values per level is shown in colored shading between a set of percentiles (75, 99, 99.5 and 99.9 %), with shading opacity increasing towards lower percentiles. The time-averaged LES profile for the period 48-72 hr is also shown as a thick solid line.

are avoided. Accordingly, the full size distribution of ice particles as detected by CIP is used. Note that the microphysics scheme does include secondary ice production, allowing the formation of large and heavy ice hydrometeors in the simulation.

Figure 10a shows that in general the measured values of liquid hydrometeor masses are situated inside the pdf of LES values. Also the simulated vertical structure seems realistic, with the largest values found near the inversion. Figure 10b indicates that the same applies to frozen hydrometeor mass, with the CIP measurements situated well within the simulated bandwidth, except for a single spike near the inversion. We conclude from these results that, apart from a few outlying samples, in general the simulated structure and amplitude of mixed-phase hydrometeor mass is realistic.

**4.5   Turbulence**

The measurements at 100 Hz of temperature and vertical velocity made by the sensors in the P6 noseboom during RF20 (Hartmann et al., 2019) allow calculating variances of both variables, as well as the turbulent heat flux. Usually a dedicated flight strategy is adopted to this purposes, featuring segments at which the aircraft flies at a fixed altitude (Nicholls and Lemone, 1980). During RF20 the southern racetrack featured five segments at constant height within the lowest 2km, as shown in Fig.





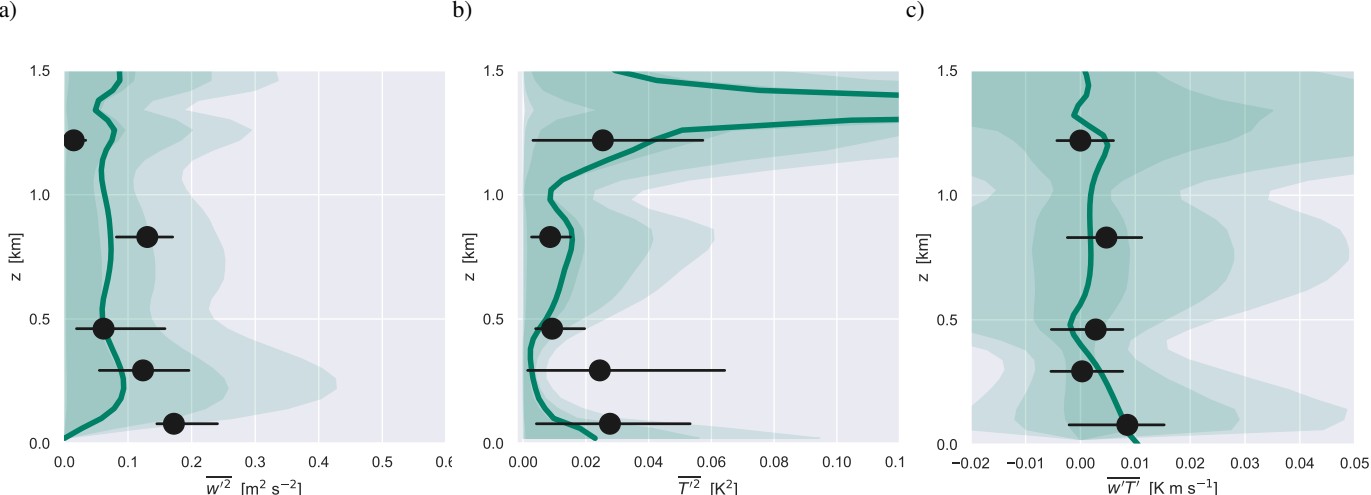

**Figure 11.** Simulated and observed turbulent (co)variances, including a) the vertical velocity variance $\overline{w'^2}$, b) the temperature variance $\overline{T'^2}$ and c) the associated temperature flux $\overline{w'T'}$. Solid black lines indicate the P6 measurements during single flight-segments at constant altitude, spanning the 5–95 % percentile interval, while the black dot indicates their median. Colored shading indicates the distribution of values encountered in LES gridboxes, with the areas between the 25-75, 10-90 and 5-95 percentiles having a decreasing opacity. The mean LES profile is shown as a thick colored line.

A1. Although this area is located slightly to the west of the DS04 target area, the near-surface turbulence is expected to be similar in both. A running average of 10 min is used to calculate a time series of the (co)variances, out of which all five segments are then lifted and interpreted.

Figure 11 shows the resulting (co)variances, here shown as a median sitting within a 5-95 percentile range of values. When comparing such turbulence data with LES it makes sense to adopt a similar approach as used in the previous section, with the

LES shown as a distribution of (resolved) values occurring at each height. Note that these distributions reflect single-gridbox values, so that extreme perturbations inside resolved convective structures (such as rising thermals) show up in the tail of the distribution. This means that again the evaluation is not quantitative, but acknowledges that the flight track might have sampled a freak event. By maintaining information about the extremes in the LES, such comparisons are still meaningful, giving insight into the probability that the measured event could also occur in the LES domain. For reference the domain

averaged (co)variances in the LES are also shown.

Both the simulation and the noseboom observations suggest the existence of a textbook turbulent mixed-layer below cloud base, featuring a concave $\overline{w'^2}$ structure, a convex $\overline{T'^2}$ structure and a linear decrease in the associated heat flux towards slightly negative values just below cloud base (at about $z = 500$ m). The latter is consistent with the local minimum in the buoyancy flux seen in Fig. 6d. At the flight segment in the middle of the cloud layer $\overline{w'^2}$ and $\overline{T'w'}$ have a second maximum, which the

LES also reproduces and reflects the impact on turbulence of latent heat release due to condensation. In general the domain averaged LES profiles are situated within the observed 5-95 percentile range, which is encouraging. The measured ranges do





occur in the LES domain for all segments, with the exception of the $\overline{T'^2}$ measurements of the $z = 300$ m flight segment of which the median clearly violates the typical concave mixed-layer structure. We speculate that this feature is caused by the strong variability in the close vicinity of convective cells, which can cause the median over a short flight segment to be larger than that of the full domain.

We conclude from these results that the amplitude and vertical structure of both intensity and transport by convection and turbulence between surface and inversion is reasonably well captured by the LES.

## 5 Results II: Joint-pdf analyses

With a satisfactory agreement between model and measurements established, we now proceed and use the numerical results to investigate the interactions between aerosol, clouds and turbulence in more detail. To this purpose we will focus on single, well-developed convective cells as appear in the last 24 hours of the simulation.

### 5.1 Spatial structure

Figure 12 digs deeper into the spatial structure of convection and clouds in the RF20 case. A convective cluster is sliced vertically, revealing moist plumes rising from close to the surface, condensing and subsequently reaching the stratiform mixed-phase cloud layer just below the inversion. This is commensurate with the cloud macrophysical structure visible in Fig. 5. Updraft speeds are substantial, with maxima at about 2 m s$^{-1}$. The updrafts are almost exclusively in liquid phase. The stratiform liquid and ice layers are deepest where the liquid updrafts connect. Interestingly, in the wider area surrounding the updrafts the top of the ice layer is also substantially higher compared to the non-convective areas, by up to about 150 m (see 12c).

More insight into the time evolution of this convective cluster is provided by Fig. 13, showing horizontal slices in the outflow layer just below the inversion at six moments during its life cycle. The cluster is relatively long lasting, slowly propagating within 1.5 hours from the bottom-right to the top-left of the domain. Starting out as a group of individual updrafts, they subsequently merge and organize into a coherent system. The perturbations in thermodynamic and cloudy state are highly collocated and correlated in the updraft core areas, showing up as narrow maxima in $w$ and $q_{\text{t}}$ and minima in $\theta_{\text{l}}$ and $q_{\text{i}}$.

An interesting observation is that often no cloud ice is present in these core areas at all. In addition, the maxima in cloud ice occur some distance away from the center of the cluster. As a result, the cloud ice has a distinct ring-like structure that surrounds the center of the cluster. This ring forms some times after cluster onset, and over time radiates out until it covers about half the domain. The ring is not present in liquid water, but does correlate with weak positive anomalies in $q_t$ and $\theta_l$. As shown in Fig. 12c), the ice ring also correlates with a slightly higher cloud top.

### 5.2 Mixing sources

While only one convective event is discussed here, these clusters frequently occur and always behave similarly. These findings motivate interpreting such clusters and their outflow area as "convective hotspots", being places where the coupling to the





**Figure 12.** Vertical cross-sections in the yz plane through a selected convective cluster present in the simulated domain. a) $q_l$, b) $w$, c) $q_i$, d) $\theta_l$ and e) $q_t$.

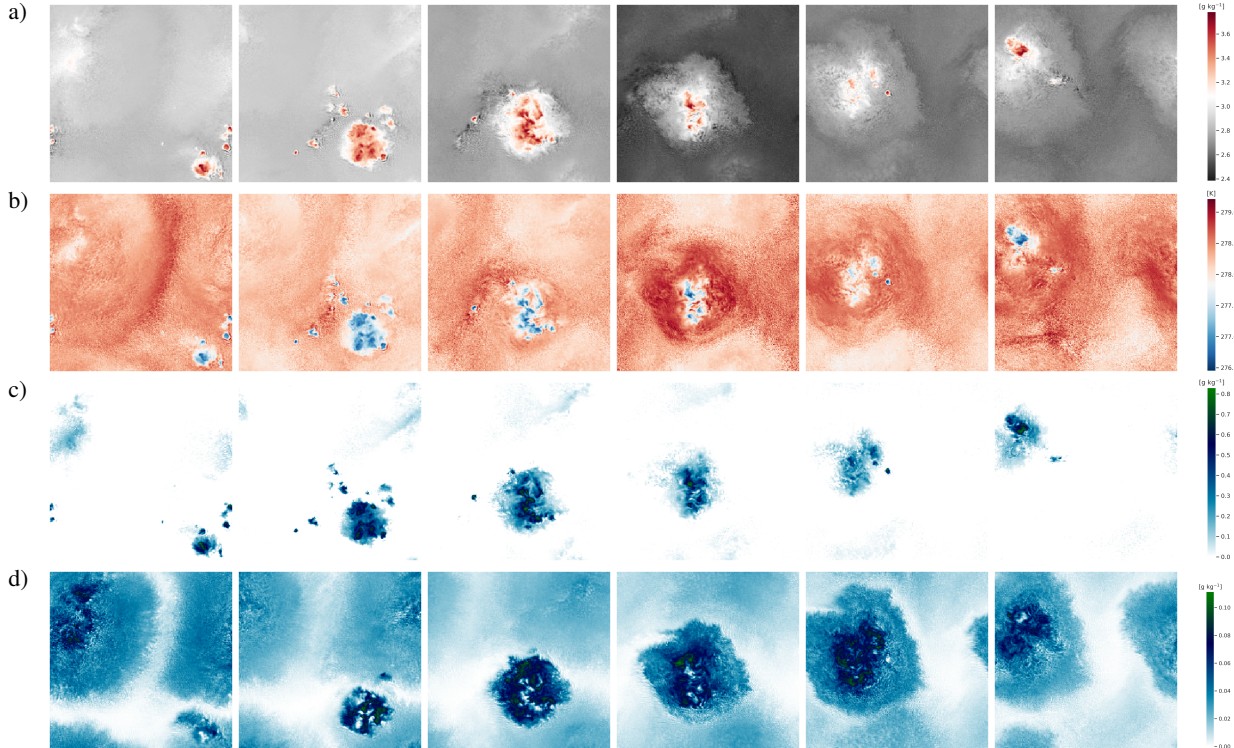

**Figure 13.** Horizontal cross-sections in the xy plane at $z = 1180$ m at six subsequent timepoints during the life cycle of a mixed-phase convective cell. a) $q_t$, b) $\theta_l$, c) cloud liquid water $q_l$ and d) cloud ice $q_i$. The time between each snapshot is 15 min. The domain size shown is $12.8 \times 12.8$ km$^2$

surface by moist updrafts is strongest, where intense hydrometeor transitions take place, and where interactions with thermodynamic state and turbulence are strongest. How these interactions work is investigated next, using joint-pdf analyses based on

scatterplots between various relevant variables. An example is shown in Figure 14, for the two-dimensional horizontal slice at the 3d time point as shown in Fig. 13. Each dot represents the state of a single gridbox on the horizontal slice. The width of these probability density functions (pdfs) represents the variance at the level of interest. In addition, the shape and orientation of the joint-pdf directly reflect the underlying physical/dynamical mechanisms that create and maintain them. These diagrams have previously been used to investigate mixing between warm convective clouds and their environment (Paluch, 1979), but

their use in investigating cold mixed-phase clouds is less established. Including hydrometeor properties in such analyses as an extra (color) dimension can provide more insight into aerosol-cloud-turbulence interactions that are unique for mixed-phase clouds.

Figure 14a shows a scatter plot of two thermodynamic state variables that are conserved for all phase changes of water, including the ice phase. These are the total specific humidity $q_t = q_v + q_l + q_i$ and the liquid-ice water potential temperature $\theta_{li}$.

In contrast to its more well-known cousin $\theta_l$ which only accounts for the latent heating of condensation $L_v$, $\theta_{li}$ also corrects





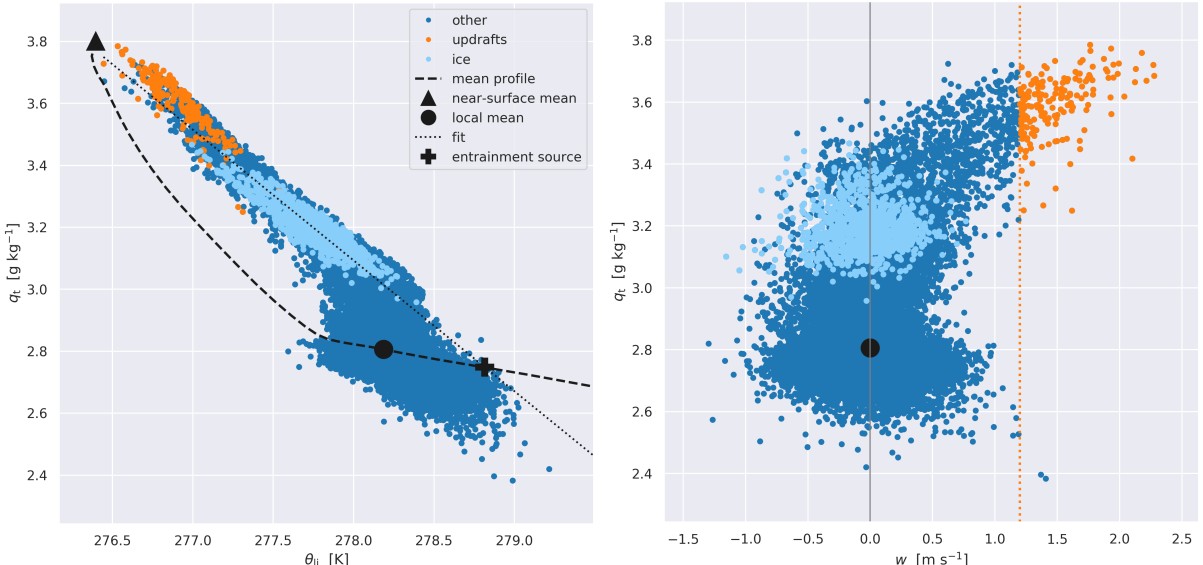

**Figure 14.** Scatterplot of various state variables at a horizontal cross-section at a selected level and time point. a) $\theta_{li}$ vs $q_t$, b) $w$ vs $q_t$. The orange points represent the strongest updrafts, defined as points meeting the double criterion $w > 1.2$ m s$^{-1}$ and $q_t > 3.2$ g kg$^{-1}$ (dotted orange line). The light blue points represent the points with most cloud ice, obeying $q_i > 0.08$ g kg$^{-1}$. The dashed black line in panel a) represents the domain-average vertical profile, with the mean states near the surface (black triangle), at sampling height (black dot, at $z = 1180$ m) and at the mixing source (black plus, at $z_m = 1289$ m) also indicated. The black dotted line is the linear fit through the mixing line, as described in the text. In b) the grey solid line indicates $x = 0$, for reference.

for the latent heating of fusion $L_i$:

$$\theta_{li} = \quad \theta - \frac{L_v q_l + L_i q_i}{c_p \Pi}, \tag{1}$$

where $\Pi$ is the Exner function and $c_p$ is specific heat capacity of air at constant pressure. In the absence of other diabatic processes, the process of turbulent mixing directly shows up in these diagrams as a tight and elongated joint-pdf, situated

between the two thermodynamic states between which the air is mixed. Figure 14a indicates that this feature, often referred to as the "mixing line", is also present in this case. Comparison to the mean vertical profile (dashed black line) indicates that the mixing takes place between moist and cold air from close to the surface (black triangle) and a point on the mean profile (black plus) that is situated just above the local mean state at this height (black circle). The presence of a well-defined mixing line in this case demonstrates that the convective mixing process in mixed-phase cloud systems acts similar to warm cloud regimes

(Neggers et al., 2002). A second cluster of datapoints can be distinguished, situated around the local mean state in a fairly horizontal direction. The widths of this pdf in both directions reflects the turbulence in the stratiform cloud layer that is driven by cloud top cooling.





Figure 14b provides more insight into turbulence by considering vertical velocity $w$. One feature immediately stands out, which is the diagonally oriented tail towards large humidity and velocity values. These are the rising moist updrafts arriving

at the inversion level, as also visible in Fig. 12. To trace these data points also in other panels such as Fig. 14a, a subset of points is now defined using the double criterion $w > 1 \, \text{m s}^{-1}$ and $q_\text{t} > 3.2 \, \text{g kg}^{-1}$ (chosen purely for visualization). These are indicated in orange in all panels. Figure 14a illustrates that all these updraft points sit in the top of the mixing line, and are associated with the strongest anomaly values in $q_\text{t}$ and $\theta_\text{li}$ at this height that are still close to the subcloud layer values (black triangle). Their high correlation indicates strong upward fluxes of heat and moisture, representing turbulent transport that acts

to maintain humidity below the inversion in the well-coupled boundary layer.

Calculating the intersection point of the mixing line with the mean profile yields the height of the air with which the convective updrafts are effectively mixing. This is achieved by least-squares fitting a line to the points with $q_\text{t} > 3.0$, which reflects the lower end of the mixing-line cluster of points. The resulting fit is shown in Fig. 14a, yielding an intersection point at the height of $z_\text{m} = 1289 \, \text{m}$ (indicated by the black plus). This height is about 100 m above the level of diagnosis, which is

consistent with earlier studies of this kind for warm convection (Blyth et al., 1988).

## 5.3  Hydrometeor transitions

The behavior of mixed-phase hydrometeors is investigated next, using the same technique. Figure 15a shows that a strong correlation exists between $q_\text{t}$ and $q_\text{l}$, reflecting that points with highest $q_\text{t}$ also have largest $q_\text{l}$. This is typically observed in moist updrafts that cool and condense while they are rising, as indicated by the orange points. The shape of the joint-PDF

of $q_\text{t}$ and cloud ice $q_\text{i}$ (see Fig. 15b) is still highly correlated but is less tight, and also has a distinct chevron shape with a tail extending towards high $q_\text{t}$ - low $q_\text{i}$ values. This tail is where the updrafts are located, as indicated by the orange points. Apparently the highest cloud ice values are not found in the rising cores; put more strongly, the strongest updrafts are almost free of cloud ice. This result is consistent with the gaps in the ice cloud field visible in Fig. 13c. Accordingly, the chevron shape seems caused by the onset of glaciation in updrafts, with the highest ice masses in the tip on the right reflecting air that has

recently experienced this transition in phase.

To gain more insight, the points that carry most ice mass are traced in these diagrams by applying the criterion $q_\text{i} > 0.08$, a value inspired by this diagram. This subset of points is indicated in light blue in all panels. While the high $q_\text{i}$ points are still situated on the mixing line (see Fig. 14 a), they are no longer rising see Fig. 14 b) and have lower $q_\text{l}$ (see Fig. 15 c). This confirms the idea that the process of glaciation mainly takes place in air that was until recently part of an updraft, turning high

$q_\text{l}$ values into ice. Geometrically, this creates a distinct ring of high $q_\text{i}$ anomalies in the radial outflow region around the updraft core, as visible in Fig. 13b.

Figures 15c-d show joint-pdfs of snow and graupel, respectively. Similar to cloud ice their extreme values are highlighted in all panels (green and purple, respectively). The strongest updrafts are almost snow-free, with the maximum snow occurrences sitting lower on the mixing line (i.e. at lower $q_t$). In that sense the snow always sits close to the cloud ice points, being their

source (see also Fig. 15a). In contrast, the highest graupel masses (purple) are highly collocated with the strongest updrafts (orange). The behavior of both snow and graupel in these diagrams reflects their formation processes, with snow forming





**Figure 15.** Same as Fig. 14 but now focusing on microphysical properties. a) Cloud liquid water content $q_l$, b) cloud ice content $q_i$, c) snow mass $q_s$ and d) graupel mass $q_g$. The green points represent extreme values of snow ($q_s > 0.03$ g kg$^{-1}$, dotted green line), while the purple points represent extreme graupel ($q_g > 0.18$ g kg$^{-1}$, dotted purple line). The black dot again indicates the mean state at sampling height.

from cloud ice and graupel generation requiring lifting by updrafts. Their preferred locations in these diagrams can thus be considered fingerprints of their genesis processes.





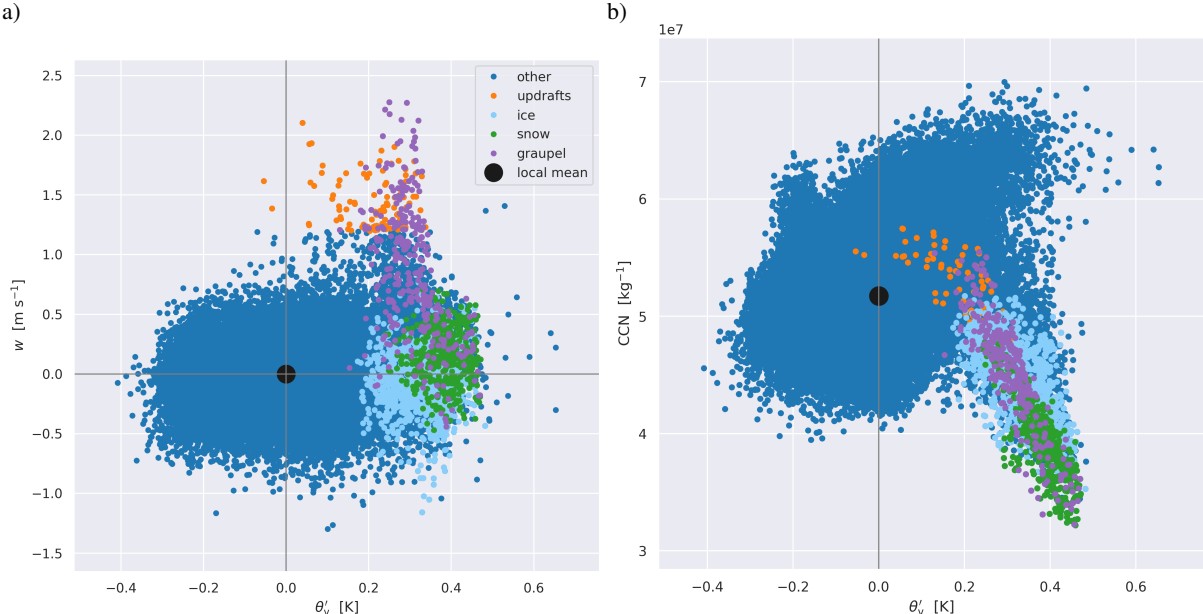

**Figure 16.** Same as Fig. 15 but now focusing on dynamics and aerosol. a) Perturbation of virtual potential temperature $\theta'_v$ (relative to the horizontal mean) versus $w$ and b) $\theta'_v$ vs CCN.

## 5.4 Dynamics and aerosol

Dynamics and aerosol characteristics are considered next. Figure 16a shows that the high $q_i$ points are relatively tightly clustered around a single point in $(\theta'_v, w)$ space, which is almost non-moving vertically but still being weakly buoyant at $\theta'_v = 0.3$ K. Taking into account that these points represent ex-updraft air that has just glaciated, the weak positive buoyancy could have been caused by the latent heat of fusion. This shared experience could partially explain why the points with most ice mass all have a similar buoyancy and are tightly clustered in this diagram, because the latent heat released is directly proportional to the

water mass that has changed phase.

Making an informed estimate can help in this respect. From Fig. 14a we estimate that, on average, about 0.4 g kg$^{-1}$ of liquid water has been converted into ice, as measured by the distance on the horizontal axis between the midpoints of the orange and light-blue sets of data points. Multiplying this by the latent heat of fusion, $L_i = 3.3e^5$ J kg$^{-1}$, and dividing by the specific heat capacity of air at constant pressure $c_p$ yields a temperature change of about 0.13 K. Comparing this to the $\theta_v$ excess of

about 0.3 K suggests that the latent heating can explain about 50 % of this temperature difference. Accordingly, we conclude that other processes must also contribute to this buoyancy level. Snow and graupel formation removes glaciated water mass, a process that also increases gridbox buoyancy. Acting together, both the latent heating of fusion and the reduction in condensate loading due to frozen precipitation formation are thus responsible for the weakly positive buoyancy in the ice ring surrounding the cluster. This positive buoyancy, which is unique for mixed-phase convection, is probably also responsible for the slightly

raised cloud top in the ice ring visible in Fig. 12c.





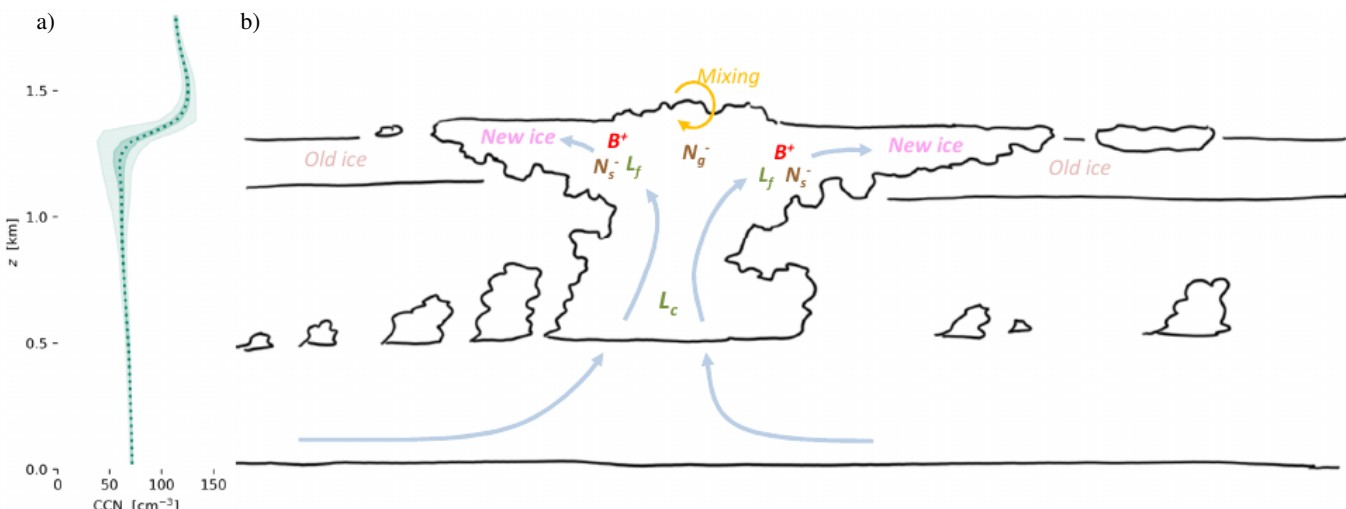

**Figure 17.** a) Vertical profile of the median of the simulated CCN at the timepoint of the joint-pdfs shown in Figs. 14 and 16b. The green shading indicates the distribution of values in the domain. b) Schematic illustration of the aerosol-microphysics-turbulence interactions in mixed-phase convective hotspots in well-coupled boundary layers over open water as identified in this study. $L_c$ and $L_f$ stand for latent heat release due to condensation and fusion, respectively. $B^+$ indicates a state of heightened buoyancy, while $N_s^-$ and $N_g^-$ indicate loss of CCN due to snow and graupel formation, respectively. Existing stratiform cloud ice is indicated as "old ice", while newly glaciated cloud ice is labeled as "new ice". Mixing between updrafts and their environment is indicated in orange. The mesoscale circulation associated with the convective hotspot is shown as blue arrows. Black lines indicate either convection-related clouds (foreground) or existing stratiform ice clouds (background).

The final component of our joint-pdf analysis concerns the behavior of CCN in the cloud layer. Figure 16b shows the joint pdf between $\theta_v'$ and CCN. Note that in this experiment the CCN is prognostic, and can change by many processes. What immediately catches the eye is the highly correlated cluster of points in the low CCN-high buoyancy quadrant. Interestingly, all three subsets of glaciated hydrometeors sit in this tail, with the snow occupying the extremest regions. This behavior suggests

that the formation of frozen precipitation is the main source of CCN depletion, with snow being the most efficient. We speculate that the strong correlation with buoyancy is probably not causal, but reflects that CCN depletion and buoyancy boosting are both driven by the same process (glaciation and frozen precipitation formation), and happen at the same location (the ice ring). This is consistent with the local minimum in the CCN profile at this height, as shown in Fig. 17a.

## 6   Discussion

While the case configuration yields a simulation that equilibrates close to the boundary layer and cloud state as observed during RF20, some aspects of the idealized experiment can have impact on the behavior of the simulated turbulence and clouds. These include the size of the turbulent domain, and the spatial and temporal discretization. The way the larger-scale




forcing is represented can also affect the results. For example, one could adopt heterogeneous forcing in a nested setup, which allows representing advection of mesoscale features into the domain that are now ignored. We speculate that this can affect the

circular structure of the convective clusters. Another important model component is the microphysics scheme, for which one expect particular sensitivity. Closely related to microphysics is the treatment of CCN and INP. In a follow-up study the authors explore some of these potential sensitivities in the LES results for this ACLOUD case, focusing on microphysics-related aspects and interpreting these in the context of the ongoing warming of the Arctic climate.

What emerges from the joint-pdf analyses is a conceptual picture of how turbulence, microphysics and aerosol interact in

well-coupled Arctic boundary layers over open water. This idea is schematically illustrated in Fig. 17b. Relatively long-lived and strong convective clusters inject moist air into the mixed-phase stratiform cloud deck. When the associated liquid clouds stop rising, they gradually glaciate while fanning outwards, creating distinct "ice rings" in the stratiform cloud layer. The associated latent heating of fusion, in combination with the formation of frozen precipitation, then cause a weak buoyancy increase, which helps to further lift the outflow cloud. The precipitation that forms, consisting of both graupel and snow, is

effective in locally removing aerosol. This probably causes the local minimum in the CCN profile immediately below the thermal inversion (see Fig. 17a), which is then mixed across the whole convective boundary layer. This vertical structure is in agreement with the ACLOUD observations (see Fig. 4), as well as other observational studies (Fitch and Garrett, 2020). The intensity of these processes is largest inside these mixed-phase convective clusters and their outflow region, which motivates interpreting them as "hotspots" or engines for enhanced aerosol-cloud-turbulence interactions in this climate regime.

# 7 Conclusions and outlook

In this study an LES experiment is performed for conditions as observed during Research Flight 20 of the ACLOUD campaign in June 2017 over open water west of Svalbard. The simulation features a mixed-phase cloud layer that is well-coupled to the surface by relatively long-lived convective clusters in liquid phase. Evaluation against in-situ and remote-sensing measurements collected by the P5 and P6 aircraft indicates that the thermodynamic, kinematic, turbulent and cloud structure are well

reproduced. Simulation output is then subjected to a joint-pdf analysis, giving insight into aerosol-cloud-turbulence interactions that take place inside the mixed-phase convective clouds. The analysis confirms that the mixing-line paradigm as well-known from warm convection also holds for mixed-phase clouds, suggesting that the mixing source of rising convective updrafts is about 100 m above them. We find that the glaciation of ex-updraft air creates distinct ring-like patterns in the stratiform ice cloud layer. The associated heat of fusion drives local buoyancy, which acts to further lift cloud top in these areas. This process

is unique for mixed-phase cloud layers fed by moist convection. Snow forms where cloud ice mass is large, while graupel predominantly forms in updrafts. Both forms of frozen precipitation further contribute to positive buoyancy in the stratiform layer. Precipitation is also efficient in locally removing CCN, creating a distinct minimum in its profile.

An important conclusion from this study is that the joint-pdf analysis in conserved variable space, best known from its previous application in studies of entrainment in warm convective clouds, can be similarly effective when applied to cold

or mixed-phase clouds. It seems particularly suited to investigate interactions between aerosol, hydrometeors and turbulent





dynamics. This motivates the further testing of this method for other cold cloud regimes, such as clouds at high latitudes and/or the tenuous cloud regime. Another follow-up activity is to find out if joint-pdf analyses of purely observational datasets, for example high-frequency in-cloud measurements of multiple variables by aircraft, exhibits the same fingerprints. This is for now considered a future research topic. We hope the results obtained in this modeling study can provide a useful context for
this effort.

The obtained results and insights also have a bearing on Arctic Amplification. One wonders how often the stagnant wind conditions as encountered during ACLOUD RF20 actually occur in the region, or more globally speaking, in areas of open water off the sea-ice edge. Such information is needed to fully understand the role in Arctic climate played by the "convective hotspots" for aerosol-cloud-turbulence interaction as identified in this study. The effectiveness of meridional transport of aerosol
and pollution into the high Arctic is affected by the occurrence of such convective events on the way. The associated overturning plays a role in the transformation of Arctic air masses, which have been identified as a key process in Arctic climate change (Pithan et al., 2018). Exploring these topics, potentially using dedicated LES experiments based on field campaign data, is for now considered a future research activity that could be inspired by the outcome of this study.

## Appendix A: P6 flight details

Figure A1 shows various properties recorded by the P6 aircraft within the southern race-track section of RF20, situated inside the target area of this study (as indicated by the orange box in Fig. 1). The section included five flight legs at constant height, used for calculating the observed covariances as shown in Fig. 11. The third panel shows the cloud liquid water content measured by the Nevzorov probe as used to evaluate the LES in Fig. 10.

## Appendix B: Composite forcings

Figure B1 shows the vertical structure of the time-constant forcings adopted for the LES experiments of ACLOUD RF20, as calculated from ECMWF analysis and short-range forecast data. Panels a) and b) show the prescribed tendencies of temperature $\dot{t}_{\mathrm{adv}}$ and humidity $\dot{q}_{\mathrm{adv}}$ due to large-scale advection. Panels c) and d) shown the zonal and meridional geostrophic wind speeds $u_g$ and $v_g$, respectively. Panel e) shows the pressure velocity $\Omega$. The shadings indicates the 1-99 and 25-75 percentile spread among included ECMWF profiles, while the median is shown as a black dotted line.

## 550    Appendix C: Microphysics scheme

### C1   Nucleation of Ice Crystals

Ice nucleation in the bulk microphysics scheme of Seifert and Beheng (2006) combines two previous approaches: Firstly, the number of activated ice nuclei is a function of supersaturation with respect to ice, as proposed by Meyers et al. (1992):

$$N_{\mathrm{IN}} = N_{\mathrm{M}} \exp\left(a_{\mathrm{M}} + b_{\mathrm{M}} S_i\right) \qquad \mathrm{if} S_i > 0 \quad \mathrm{and} \quad T < T_{\mathrm{M}}, \tag{C1}$$





**Figure A1.** Timeseries data from the P6 flight RF20 during its southern race track section. a) Longitude, b) altitude above sea level, and c) cloud liquid water content $q_l$ as measured by the Nevzorov probe onboard P6. The fixed-height flight segments used for covariance calculations in Section 4.5 are indicated in black.

where $T$ is the absolute temperature, $S_i$ is the supersaturation with respect to ice surface, and the values of parameters follow: number parameter $N_M = 10^3 \, \text{m}^{-3}$, the intercept coefficient $a_M = -0.639$, and the linear coefficient $b_M = 12.96$, and finally the threshold $T_M = 268.15 \, \text{K}$ limits below which temperatures does the ice nucleation occur. Secondly, in order to avoid very low number concentrations, the nucleation is limited to be within one order of magnitude from the modified Fletcher's formula (Fletcher, 1962; Reisner et al., 1998):

$$N_{IN,F} = 10^{-2} \exp\left(0.6 \left(T_0 - \max(T, T_{min})\right)\right) \qquad \text{if} \quad T < T_M, \tag{C2}$$





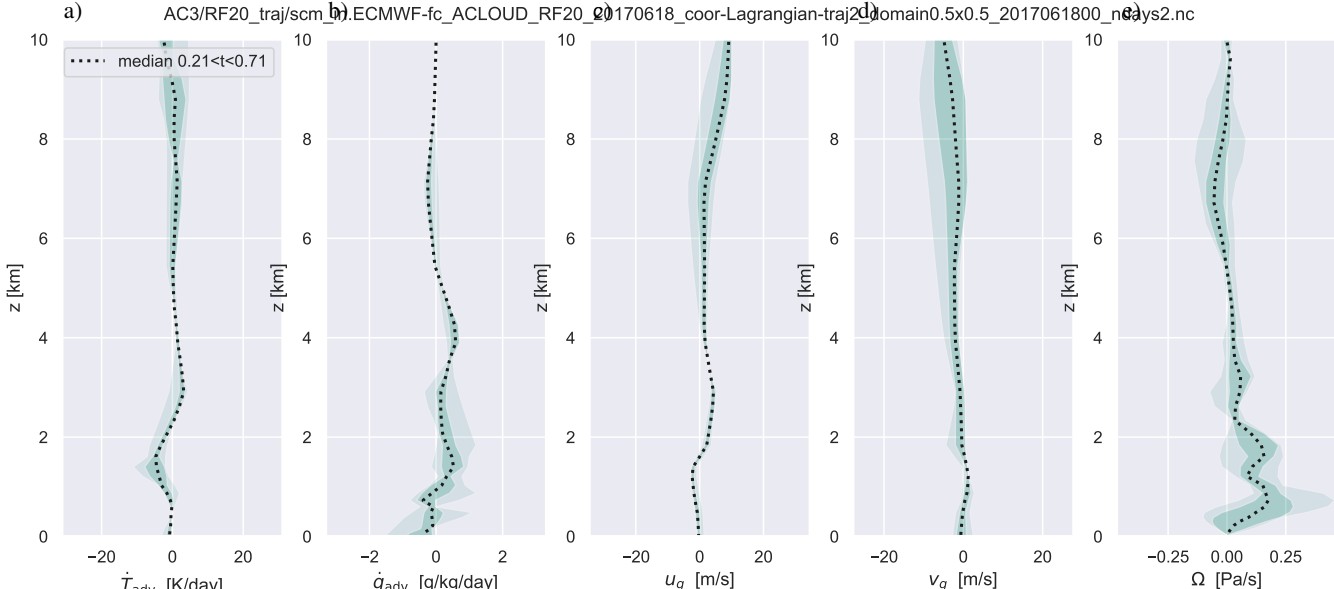

**Figure B1.** Profiles of the prescribed large-scale forcings based on ECMWF data.

**Table C1.** Overview of setting microphysical parameters for hydrometeors

|  | $a$ | $b$ | $\alpha$ | $\beta$ | $\gamma$ | $\nu$ | $\mu$ |
|---|---|---|---|---|---|---|---|
|  | $(\mathrm{m\,kg^{-b}})$ |  | $(\mathrm{m\,s^{-1}\,kg^{-\beta}})$ |  |  |  |  |
| cloud droplets | 0.124 | 1/3 | $3.75 \cdot 10^5$ | 2/3 | 1 | 1 | 1 |
| raindrops | 0.124 | 1/3 | 159.0 | 0.266 | 1/2 | −2/3 | 1/3 |
| cloud ice | 0.217 | 0.302 | 41.9 | 0.36 | 1/2 | 1/3 | 0 |
| snowflakes | 8.156 | 0.526 | 27.7 | 0.216 | 1/2 | 1 | 1/3 |
| graupel | 0.190 | 0.323 | 40.0 | 0.230 | 1/2 | 1 | 1/3 |

The overview covers the setting of microphysical parameters for the size and velocity of hydrometeors, as well as
for particle mass distribution of hydrometeors under the assumption of generalised gamma distribution.

where $T_0$ is the freezing point of water and $T_{\min} = 246\,\mathrm{K}$ limits production at extremely low temperatures.

## C2    Freezing of Hydrometeors

The timescale in the heterogeneous freezing of supercooled cloud droplets is dependent on their size and temperature, following
Pruppacher and Klett (1997) and Khain et al. (2000). The timescale in the homogeneous freezing of cloud droplets is then





given by Cotton and Field (2002). The heterogeneous freezing of raindrops is analogous to the heterogeneous freezing of cloud droplets. The only difference lies in classifying the resulting particle as graupel.

## C3  Secondary Ice production

The secondary ice production includes ice multiplication by Hallet-Mossop process, occurring during the riming of ice hydrometeors in the temperature ranges between 265 K and 270 K (Hallett and Mossop, 1974; Griggs and Choularton, 1986). The number of ice splinters released during the process is dependent on the temperature and the riming rate, following the parameterization of Beheng (1982).

## C4  Nucleation of Cloud Droplets

The nucleation of cloud droplets again follows Seifert and Beheng (2006) as closely as possible. Firstly, the nucleation rate is calculated explicitly as

$$\left.\frac{\partial N_c}{\partial t}\right|_{\mathrm{nuc}} = \widetilde{C_{\mathrm{ccn}}} \ \kappa \ S_l^{\kappa-1} \ w \ \frac{\partial S_l}{\partial z} \qquad \text{if} \quad S_l \geq 0, \ \text{and} \ S_l \leq S_{\max}, \ \text{and} \ w \ \frac{\partial S_l}{\partial z} > 0 \tag{C3}$$

where $\widetilde{C_{\mathrm{ccn}}}$ and $\kappa$ are CCN parameters, $S_l$ is supersaturation with respect to liquid water surface (expressed in %), $w$ is the vertical velocity of the air, and $S_{\max}$ is a threshold for saturation when all available CCN are activated. Secondly, nucleation rate is limited so the number of droplets does not exceed $N_{\mathrm{ccn}}$, the current CCN number concentration. While RF20 reflects maritime conditions, the values of $\kappa$ parameter and the saturation threshold are set to constant values $\kappa = 0.462$ and $S_{\max} = 1.1\%$ (Seifert and Beheng, 2006). Unlike in the original description, the parameter $\widetilde{C_{\mathrm{ccn}}}$ is not the constant $C_{\mathrm{ccn}}$, but it is instead dependent on the aforementioned variable $N_{\mathrm{ccn}}$. The consistency with the power law relation for activation spectra is maintained by calculating this parameter as

$$\widetilde{C_{\mathrm{ccn}}} = \frac{1}{\left(S_{\max}\right)^\kappa} \ N_{\mathrm{ccn}}. \tag{C4}$$

The values of other important microphysical parameters are shown in the Table C1.

*Acknowledgements.* We gratefully acknowledge the funding by the Deutsche Forschungsgemeinschaft (DFG, German Research Foundation) – Projektnummer 268020496 – TRR 172, within the Transregional Collaborative Research Center "ArctiC Amplification: Climate Relevant Atmospheric and SurfaCe Processes, and Feedback Mechanisms $(\mathcal{AC})^3$". We thank ECMWF for providing access to the large-scale model analyses and forecasts fields used to drive the LES experiments. We gratefully acknowledge the Regional Computing Centre of the University of Cologne (RRZK) for granting us access to the CHEOPS cluster. The Gauss Centre for Supercomputing e.V. (www.gauss-centre.eu) is acknowledged for providing computing time on the GCS Supercomputer JUWELS at the Jülich Supercomputing Centre (JSC) under projects CHKU28 and VIRTUALLAB. We further thank the Alfred Wegener Institute (AWI), the PS106/1 crew and the ACLOUD science teams for making the field campaign happen and for post-processing the observational data.



*Code and data availability.* The modified version of DALES with extension for mixed-phase microphysics with interactive CCN (dales4.3_sb3)
used in this model study is marked as release https://doi.org/10.5281/zenodo.5642477 and for the licensing information please see the main
DALES GitHub page: https://github.com/dalesteam/dales. The files containing the ACLOUD RF20 case configuration, as well as the main
model output, are available at https://doi.org/10.5281/zenodo.5642505 (last update October 2021).

*Author contributions.* SM provided the aerosols field measurements, as well as the guidelines on the treatment of aerosols. RN designed the
model framework and strategy, and prepared the model forcing files. JC developed the model extension for the interaction of aerosols and
600 mixed-phase microphysics. MM operated the radar and lidar and performed measurements. BK provided the lidar data. MM and BK adviced
on visualisation and interpretation of remote sensing data. JC collected the ACLOUD observational datasets, performed the model simulations
and processed the output. RD provided guidelines on the treatment of in-situ cloud measurements. CL and DC provided guidelines on the
other airborne instruments, as well as the description of the field campaign. JC and RN worked together on analyzing the results and
evaluating them against the measurements. JC prepared the manuscript, of which RN revised various intermediate versions.

*Competing interests.* The authors declare that they have no conflict of interest.





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
