# Peer review of "Aerosol impacts on the entrainment efficiency of Arctic mixed phase convection in a simulated air mass over open water"

_Atmospheric Chemistry and Physics, 2021_

## Referee Comment (RC1)

**Review** of a manuscript "Aerosol-cloud-turbulence interactions in well-coupled Arctic boundary layers over open water" by Jan Chylik, Dmitry Chechin, Regis Dupuy, Birte S. Kulla, Christof Lüpkes, Stephan Mertes, Mario Mech, and Roel A. J. Neggers, considered for publication in ACP.

**Overall recommendation**: publish after major revisions.

This manuscript discusses high-resolution (LES) numerical simulations of an Arctic cloud system observed during the ACLOUD campaign. Overall, I see significant problems with the manuscript and feel considerable revisions are needed before the paper is accepted. Overall, I feel the paper does not present any scientific hypothesis, and the main goal seems to be to demonstrate that the model is capable in reproducing observed cloud features. If this is indeed the case, the presentation does not reach that goal. Below I present my main points and then follow with numerous specific line-by-line comments. Addressing all those should lead to a publishable manuscript.

**General (major) comments**

1. Manuscript title should have "modeling" word in it. Perhaps start with "Modeling of aerosol-cloud…". That said, aerosol-cloud interaction aspect of the presentation is questionable to me, see 3 and 4 below.

2. Figure 1 in the manuscript documents presence of complex cloud systems in the area. Is there anything there that resembles the cloud patch simulated by the model? Perhaps a couple figures showing how clouds evolved in time (to ensure the way they are simulated makes sense) would be useful. If this is not possible (i.e., not possible to show that similar structures persisting for a few days), then what is the general motivation behind LES simulations? I understand that having a periodic relatively small LES domain imposes stringent limitations, but these must be at least mentioned when compared natural clouds and simulated virtual reality.

3. I have serious doubts about the mixed-phased microphysical scheme used in the simulation. For warm rain, the scheme applies saturation adjustment. What are simulated and observed droplet concentrations? I assume there are some aircraft observations to consider for simulation validation. For ice, how important is the saturation adjustment that warm rain scheme uses? For instance, sizes of cloud droplets affect ice initiation. Are ice concentrations in general agreement with aircraft observations? Only mass of water and ice are shown in the paper. Also, satellite retrievals should be helpful here as well. Maybe even partitioning between liquid and ice can be obtained from the satellite. Overall, there are much better double-moment mixed-phased schemes available today (not to mention bin microphysics) that should allow more confident simulation of microphysical process. Limitations of the scheme applied in this study needs to be at least mentioned in the manuscript.

4. Following 3 above, it is unclear how CCN and IN are prescribed and how they evolve in the simulation. This is important for specific results, see specific comments below.

5. When thinking about the simulated case, I wonder to what extent presence of ice in the system is important. From the basic dynamics point of view, I think ice does not matter. Maybe it does for the radiative transfer, but this is not obvious. Such thinking might be along the lines of a hypothesis that modeling can address. That said, my intention is *not* to ask the authors to run a simulation without ice, but just to suggest something for a future investigation.

6. I find the technical quality of the presentation relatively low. Some figures require adjustments to make them legible when included in the printed version. See specific examples below. Some statements in the text require revisions, again, see specific examples below. Overall, the conclusion section is relatively short and perhaps reflects on the rather "thin" outcome of this study, at least as discussed in the current version.

**Specific minor comments**

1. L. 48: "primitive equations" are typically referred to as the set of mesoscale hydrostatically-balanced equations. Please rephrase.

2. L. 127: "…our main objective is to study interactions….". This does not seem a correct statement. Yes, there is a simulation, but the interactions are not studied. And I am not sure what would be needed to make this statement correct. Sensitivity simulations with different CCN and IN concentrations? Different ice initiation mechanisms? More convincing comparison with observations?

3. Figure 4. Because the distributions overlay, I think it would be better to plot distributions in two panels. Also, is the information about solubility available (i.e., do all can serve as CCN?). Why such a large difference for aerosols above and below cloud layer? Large-scale advection (for above) and surface emission (for below)?

4. L. 172. "upwind scheme". Really? I do not think so. If this is correct, then this is a serious problem.

5. Section 3.2. Is there any justification for specific parameter values selected for the simulation? CCN and IN concentrations in particular. Is the way ice is initiated in the model justified?

6. L. 238. I assume there is longwave radiative transfer as well. Please mention details.

7. L. 251: "Radiation is interactive…" is an incorrect statement. I think this refers to the fact that all hydrometeors are seen by the radiation code. Perhaps some details would be need here.

8. L. 256/257. "Continues nudging" and "Newtonian relaxation". Please use one term to avoid confusion. Also, such an approach dumps the perturbations, correct? Within the boundary layer, one can still relax the *mean* towards the observations (i.e., no dumping small-scale perturbations), but I think this not used, correct?

9. L. 265: "CCN … can evolve freely". Really? This means that with precipitation the CCN (and perhaps IN) are being removed. The text further in the draft confirms that. Is this a fair model setup, that is, with changing in time CCN? Should the concentration be maintained? What impact the reduced CCN later in the simulation has? I would expect some differences, for instance, in the rain mass.

10. Fig. 4 should include panel boundaries. The text to the right of each panel is too small and will be even smaller in print. Please change. L. 288 mentions diurnal cycle. I do not see it in the panels.

11. L. 303. "… simulated profiles agree well with observations…". The relaxation ensures that, at least for some variables, correct?

12. Fig. 9. The text explains how the figure is created for the model output. What about the observations? What is the difference between mirac and lidar? Can they distinguish water from ice? Overall, the figure rises more questions and it suggests a poor comparison. What is the reason for showing it?

13. Fig. 10. As in 12 above: the comparison is poor. I understand that it is difficult to compare model output averaged in time with a small number of aircraft legs and a few heights, but what do we learn from the figure? And why only mass? I would think considering particle concentrations and sizes would be more informative (e.g., per my points above).

14. Fig. 11. Please include panel boundaries. Zero line would help in panel c. How do data from the aircraft are processed (i.e., averaged over what distance)? Do the figure (and Figs. 9 and 10) allow meaningful comparison between model and observations? I am not sure. More details are needed.

15. L. 366. Please provide some references in support "textbook" boundary layer and cloud structures.

16. Section 5.1, Fig. 12 needs panel boundaries. L. 392: should it be "left to right"? Fig. 13: labels on the color scale are too small. Looking at the panels, the question about the domain size comes to mind. Are there any satellite observations that would support the simulated presence of mostly water in the updraft region and ice at the peripheries? Does that partitioning depend on microphysical scheme assumptions?

17. Fig. 14 and reference to Paluch (1979). The analysis is strictly valid for nonprecipitating clouds, correct? Does that explain larger scatter of data points in the upper part (i.e., lower right corner of the left panel)? L. 413 and eq. 1: The liquid-ice potential temperature (as well as liquid water potential temperature) is not conserved when precipitation is present. L. 427: radiative cooling and fallout of precipitation. The caption to Fig. 14: "at a selected level and time point": please provide details. Does the figure change if a different level and time is selected? Similar comments apply to Figs. 15 and 16. Please explain.

18. Figs 15 and 16. Please explain the height the data come from. Overall, how Figs. 14 – 16 are created should be explained. Symbols and number along the axes as well as symbols inside should be larger. Besides just discussing what the figures show, what are physical outcomes of the analysis? I feel the discussion is quite thin in that respect. Some of the points made in section 5.4 are difficult to follow without details on how aerosols are treated in the microphysics scheme.

19. With all the comments above, I have not read sections 6 and 7.

---

## Referee Comment (RC2)

Review of ACP-2021-888: "**Aerosol-cloud-turbulence interactions in well-coupled Arctic boundary layers over open water**" by Jan Chylik, Dmitry Chechin, Regis Dupuy, Birte S. Kulla, Christof Lüpkes, Stephan Mertes, Mario Mech, and Roel A. J. Neggers

Ian Brooks

**Overview**

This paper examines a large eddy simulation of a convective cloud system in the Fram Strait, based on and validated against airborne measurements from the Polar 5 and Polar 6 aircraft. It is a not untypical case for the marine environment on the margins of the Arctic sea ice, but perhaps less studied than the near-neutral boundary-layer capped by stratus/stratocumulus that dominates over the sea ice. Given the rapid warming of the Arctic and increasing loss of sea ice, the conditions represented here are arguably likely to be more prevalent over a wider area within the Arctic in the future, motivating a more detailed study.

The nominal aim of the paper, encapsulated in the title, is a study of 'aerosol-cloud-turbulence interactions'; the analysis of these interactions is, however, rather cursory and for the most part does little to push forward the scientific understanding of these convective cloud conditions. The analysis of aerosol interactions really does little more than demonstrate that the model's CCN sinks (snow and graupel formation) do indeed remove CCN.

I get the strong impression from mentions of planned or potential future studies, that this paper is primarily intended as a description of a baseline case (the only model run discussed is referred to a a 'control simulation'), demonstrating that the model successfully represents the observed conditions, with the real science – process sensitivity studies, etc – expected to be covered by future papers. In that case the earlier parts of the paper (notably section 4: 'Evaluation') covering the basic validation against the aircraft observations, might be better presented as supplementary material to a paper more strongly focused on a scientific question.

In many places the details required to properly assess the data visualised is not given, e.g.: times of model outputs, periods over which averages/PDFs are generated. Specific cases are noted below. A good paper should provide enough information for a reader to repeat the analysis – we might be able to download the data, but are not always given sufficient information to know which bit of it is needed to reproduce a plot.

The analysis focuses on a single convective cluster, located within a region of about ¼ of the larger domain, and over a short period of time (~1.5 hours). While it is not unreasonable to assume that the behaviour of this convective cluster is representative of such features in general, we are not given any evidence to support that assumption. Is this the only region of convection over this time period? If so, then the choice of what portion of the domain to assess variables over may bias the results since the convection takes up a larger fraction of the sub-domain than of the whole domain. Similar issues pertain to the time averaging…how often do such convective events initiate? Are the results presented truly representative of the wider environment, or biased by the selection of one event? There is much to be learned from case studies of single events, but it they give no idea of the natural variability of such systems, and it may be misleading to try and generalise from them.

The analysis in figures 14, 15, and 16 is informative, but far more could be revealed with conditional sampling. For example in 14a, three points representing layer means are shown. Those for the near-surface and 'entrainment source' are well defined (although we can't tell where the entrainment

source actually is in the vertical), but the mean for the layer from which all the individual points are taken has a value that is likely to depend on the ratio of cloud to clear-air. It would be instructive to also see the means for cloudy and clear air, with the latter being more representative of the background environment in which the convection is embedded.

Similarly, a lot could be learned from conditional sampling of points in updrafts, downdrafts, by liquid or ice content (beyond the simple extremes already indicated). And by looking at the time evolution of the system.

I am left rather frustrated by what is clearly a very extensive LES simulation by a very capable model, but for which the analysis presented is very limited. I'm not sure what, of the results shown, would actually be worth citing other than by future studies using this model and base case, as a demonstration that the model accurately represents the observations.

**Detailed comments**

Line 30-31: The statement "A large variety of cloud forms occurs at high latitudes; one possible way of classifying these can be based on the properties of the underlying surface" mixes up different things. The surface below the clouds is more relevant to the boundary layer environment within which the clouds exist than as a means of classifying the clouds.

The authors then define 3 regimes (pack ice, marginal ice zone, and open water) and then state that only the open water case will be studied. The whole bit about cloud forms and classification regimes seems rather unnecessary – as if they feel it necessary to justify looking at clouds over open water rather than ice.

Line 48: "LES stands for the numerical simulation of the discretized set of primitive equations for geophysical flow…" – I know what the authors mean here, but the words 'stands for' give the wrong impression. 'LES' stands for (ie is an acronym for) Large Eddy Simulation…which is all ready established. "LES represents…" or "LES models perform a numerical…" might better convey the intended meaning.

Lines 80-86: there's quite a bit of repetition here from the previous section. Recommend providing the details in one or the other, but not both.

Line 97-98: "The air mass in the Fram Strait was slow-moving on this day, being situated over relatively warm open water." – the phrasing here, 'being situated…' implies a causal link: that the air mass was slow moving *because* it was overlying warm water. That is not the case, change 'being' to 'and'.

Figure 7: the shading in the plots, representing the spread in values, are very narrow. It really isn't worth including the contoured 5,25,50,75,95% intervals – the really can't be seen, and provide no useful information here.

Figure 9, and line 328-331: what time periods are LES PDFs of cloud top height generated over? Is this a single time or aggregated over many output times?

Figure 11, and line 360: "LES shown as a distribution of (resolved) values occurring at each height. Note that these distributions reflect single-gridbox values" – it isn't clear exactly what is meant here. Do these profiles represent the mean (and shading the percentile ranges) of all the grid boxes at a specific altitude at one time, of a single column through the model over a period of time, or of all the grid points on a level over multiple output times? If there time averaging, over what period?

Figure 11, lines 366-367. The authors state "Both the simulation and the noseboom observations suggest… featuring a concave $w'^2$ structure, a convex $T'^2$ structure…", this doesn't seem to match the figure. The $w'^2$ results below ~250m are rather different for LES and obs, with the LES decreasing towards the surface while the observations increase to a maximum at their lowest altitude (~50m). The curvature of the observations below 500m appears convex for both $w'^2$ and $T'^2$ contradicting the statement, and for $T'^2$ the LES and observations have opposite curvatures, not the same, as implied by the text.

Then at line 372 the text claims the $T'^2$ profile has a concave structure, contradicting the previous statement.

Figure 12. At what time are the vertical cross sections obtained?

Figure 13: The colorbar text is far too small – I can't read either the variable units or scale values without zooming in a long way. (same applies to a less extent on figure 12).

Line 391, Figure 13: at what times are the model outputs obtained? We're told they are at 15 minute intervals, but no more. How do these times relate to the time of the vertical cross sections in figure 12? I assume (hope) on of them is at the same time.

It would be useful here if the vertical level of the figure 13 horizontal cross sections were indicated on figure 12 to help put them into context. And similarly if the location of the vertical cross sections in fig 12 was indicated on figure 13 (maybe just on the panels from the same time…assuming they are at the same time).

Line 394: the text referring to figure 13 states "…showing up as narrow maxima in w and $q_t$ and…" but 'w' is not plotted in figure 13.

Line 397: "This ring forms some times after cluster onset, and over time radiates out…" how long after cluster onset? How long does it take to move outwards? The lack of any information on temporal evolution of this system is a weakness of the analysis.
Also 'times' should be 'time'

Line 404 and elsewhere: The plots in figures 14, 15, and 16 are referred to throughout the text as representing 'joint-pdf' analyses. These are not joint-PDFs, there are no probability distributions presented here. These are simple scatter plots of individual data points. Figure 14a is formally a mixing diagram (as per Paluch, 1979) since the variables plotted are conserved under adiabatic processes and phase changes of water. The remaining panels in all three figures are simply scatter plots rather than mixing diagrams, since at least one variable in each case is not a conserved quantity. This distinction is not made clear, and confusion might ensue for a reader unfamiliar with this, since significant discussion is made of 14a as a mixing diagram, and then the caption for figure 15 states 'Same as Fig. 14', which is true in that they are both scatter plots, but misleading if one thinks of fig 14 as a mixing diagram.

Line 406: "the 3d time point" – should this be "the 3rd time point"?

Line 407: "The width of these probability density functions (pdfs) represents the variance at the…" – no, the width of the scattered points represents the full range of data values. We cannot judge how many points might cluster on top of each other near the centre of the distribution – the more of them there are, the smaller the variance. We need an actual PDF to be able to judge the variance.

Figure 14 caption: "Scatterplot of various state variables at a horizontal cross-section at a selected level and time point…" – what time? Which point(s)?

Line 422: "and a point on the mean profile (black plus) that is situated just above the local mean state at this height (black circle)" this is…unhelpful. What is meant by 'just above'…how far above. what constitutes 'just above'...how far is that?

The figure legend indicates that the black + is the "entrainment source" and the caption that it is the "mixing source". 'Mixing source' doesn't help identify *where* it is, and while 'entrainment source' implies a point within or just above the inversion, exactly where, and how is this level selected? Give a specific, quantitative definition.

Line 425: "A second cluster of datapoints can be distinguished, situated around the local mean state in a fairly horizontal direction." – 'in a fairly horizontal direction' is rather vague, what is 'horizontal' on a plot so $q_t$ vs $\theta_{li}$? You mean that it has a wide range of $\theta_{li}$ while clustering around a narrow range of $q_t$.

Line 426: "The widths of this pdf in both directions reflects the turbulence in the stratiform cloud layer" – the range of values of both variables represents something of the mixing history, but not necessarily of active turbulence. If turbulence stopped, this variability would remain.

Line 439: "This height is about 100 m above the level of diagnosis" – what is the 'level of diagnosis'? Do you mean the level at which the horizontal cross section is taken?

Line 447: "Apparently the highest cloud ice values are not found in the rising cores" – we already know this, it was observed in figure 13.

Line 449: "Accordingly, the chevron shape seems caused by the onset of glaciation in updrafts," – isn't this inconsistent with the updrafts being 'almost ice free'?

Line 454: "This confirms the idea that the process of glaciation mainly takes place in air that was until recently part of an updraft" – we might infer this, but without information on the time evolution of the system it is perhaps rather overstating the case to say it 'confirms' this.

Line 459: "the snow always sits close to the cloud ice points, being their source" – the grammar inverts the causal relationship, as stated this implies the snow is the source of ice, rather than the other way around.

Line 466: "which is almost non-moving vertically" – yes the centre of the distribution is at w ~=0, but it spans the range ±0.5 m/s with outlies to ~±1 m/s – so most individual grid points do have an appreciable vertical motion. Typical for a turbulent layer.

Line 477: "Snow and graupel formation removes glaciated water mass, a process that also increases gridbox buoyancy" – having estimated the buoyancy contribution from latent heating, could you not also estimate the contribution from snow and graupel formation? The model presumable outputs all the terms in the heat budget, so you could find all the fractional contributions to the buoyancy perturbation.

Line 483: "the highly correlated cluster of points in the low CCN-high buoyancy quadrant. Interestingly, all three subsets of glaciated hydrometeors sit in this tail, " – this is exactly what you expect given that the model is formulated so that formation of snow and graupel is a sink of CCN. I'm not sure that this counts as a result.

Line 485-487: "We speculate that the strong correlation with buoyancy is probably not causal, but reflects that CCN depletion and buoyancy boosting are both driven by the same process (glaciation and frozen precipitation formation), and happen at the same location (the ice ring)." – this could be examined by conditionally/spatially subsampling the domain.

Line 516: "The analysis confirms that the mixing-line paradigm as well-known from warm convection also holds for mixed-phase clouds" – doesn't it have to, it is a fundamental property of conserved quantities representing heat and moisture.

Line 517: "suggesting that the mixing source of rising convective updrafts is about 100 m above them" – this is a bit vague. What is meant by 'mixing source', and 'about 100m above them' (ie above convective updrafts) is very vague...what is your definition of the top of the updraft from which that '100m above' starts?

Line 525: "It seems particularly suited to investigate interactions between aerosol, hydrometeors and turbulent dynamics" – this is a bit misleading. 'It' refers back to the subject of the previous sentence, which was conserved variable mixing diagrams, but none of the variable here are conserved. Yes scatterplots can be informative here, but they are not the same as mixing diagrams.

Line 531: "One wonders how often the stagnant wind conditions as encountered during ACLOUD RF20 actually occur in the region,…" – I appreciate this is outside the scope of this study, but there are archives of reanalysis data which would answer this question.

Figure B1. The overlapping titles on the panels make them more or less illegible. Move this information into the figure caption.

---

## Author Comment (AC1)

**Final author comments (ACs) for manuscript acp-2021-888**

**General response to all reviewers**

We would like to thank the reviewers for carefully reading and assessing our submitted manuscript. We conclude from these reviews that the LES simulations we conducted of the boundary layer clouds observed during ACLOUD RF20 are in principle worthwile, and have scientific value. We also appreciate the positive comments on the basic configuration of the LES experiment, and its evaluation against the observational datasets.

However, both reviewers also have major concerns, which need to be addressed before publication is possible. In our understanding, the most serious issue seems to be the perceived lack of a clear scientific hypothesis and associated analysis that is worth citing, that goes beyond just showing that the LES reproduces the observations. Or in other words, that the scientific outcome is "too thin". A second major concern is the technical and scientific quality of the presentation, which was judged to seriously lack at various points.

To adequately address these concerns we propose to introduce substantial changes in the manuscript. Firstly, we would like to adopt a different main science question or hypothesis, which is that the aerosol concentration in an Arctic air mass significantly affects the efficiency of radiatively driven top-entrainment in heating the convective boundary layer. The new analysis will thus replace the mixing-line and scatter plot analysis that was included in the first submission (Section 5 named "Results II: Joint-pdf analyses"), which received considerable criticism in the review. Note that the basic evaluation of the control LES experiment against observational datasets from ACLOUD will be maintained.

The new hypothesis is motivated by the following points. Firstly, it can well be tested with the currently adopted experimental setup. Secondly, while the efficiency of entrainment in heating the boundary layer has been studied for subtropical warm stratocumulus clouds (Stevens et al., 2005), this is not yet the case for Arctic mixed phase clouds. Thirdly, ongoing work on this LES case by the authors since submitting this manuscript has yielded some new insights into this topic. We find that the heating by radiatively driven entrainment plays a key role in the heat budget of the lower atmosphere, as well as the surface energy budget. Sensitivity experiments we conducted also show that the CCN content of the Arctic boundary layer significantly changes the entrainment efficiency, affecting the lapse rate at lower levels.

The heat budget and lapse rate play a crucial role in Arctic Amplification, making this behavior relevant and of interest to the community. To our knowledge these insights are novel, and have not been published before. We hope that the new focus, hypothesis, associated analysis and results add the extra value to the paper that was asked for, and will make it worth citing. Apart from these major changes, we also tried hard to improve the presentation of the results and the quality of the figures. In combination, we hope these changes can address all concerns of the reviewers.

A detailed response to all comments by both reviewers is provided below.

**Response to Reviewer 1**

We are glad to read that the reviewer in principle finds the manuscript publishable, pending a few major revisions. Briefly summarized, the main issues identified by the reviewer include the following:

1. No scientific hypothesis is presented

2. The main goal seems to be to reproduce the observed clouds

3. The presentation does not reach that goal

The substantial changes in the revised manuscript as already described above in our general response to all reviewers should be sufficient to address the first point. These changes were also motivated by the comments of the second reviewer. To briefly summarize, the new focus is the heating efficiency of radiatively driven entrainment in mixed phase clouds, with the associated hypothesis that this process is significantly affected by CCN. The new focus and hypothesis are now also described more clearly in the introduction. Concerning the second issue, we now state more clearly and exactly what we expect from the model evaluation against the observations. We hope we can thus avoid raising the wrong expectations. The evaluation method is similar as applied in previous LES studies of this kind. Finally, to address the third point we put great effort into improving the presentation, both in content and technical quality.

Overall, we hope that these adjustments are sufficient for addressing the main concerns of the reviewer. We are looking forward to the assessment of the revised manuscript.

*Response to general (major) comments*

1) Modeling not in title / Aerosol-cloud interaction aspect is questionable

As requested, we modified the the title to include the word "simulation".

The aerosol-cloud interaction that is examined in this study is now specified more accurately: the impact of aerosol on the heating efficiency of radiatively driven entrainment in Arctic mixed-phase clouds.

2) Complex cloud systems / Time evolution of clouds / General motivation behind LES simulations

A detailed evaluation of the geometry and time evolution of individual cloud structures in the simulation is not the objective of this study. Like most previous LES studies of Arctic clouds, the evaluation of model performance is mainly done in a statistical way. That means we seek agreement on bulk properties such as cloud mass, phase and height. This is sufficient for showing that the control LES run reproduces the observed mean structure of the observed boundary layer and clouds, and that it can serve as a reference experiment in further sensitivity runs.

Recent LES studies of this kind include Ovchinnikov et al. (2017) and Stephens et al. (2018), in which multiple LES codes took part. Often the lack of observational data is a problem in such studies. The in-situ cloud data from ACLOUD RF20 already allows us to perform a much more thorough evaluation. However, a detailed evaluation of individual cloud geometry and life-cycle would require additional data that were not collected during the campaign.

To avoid raising too high expectations, in the revised manuscript we explain in more detail what we expect from the evaluation, and also put our study in the context of previous LES studies of Arctic clouds. We also express more clearly in the introduction what is the motivation of doing LES of this case: i) an understudied cloud regime, ii) a key area for air mass transformations by low level

convection, iii) availability of unprecedented measurements, and iv) opportunities for defining a representative control case for impact studies of CCN on entrainment.

Concerning the time evolution, it should be noted that we use a composite case setup. This means that forcings are time constant, and reflect a time average. As a result, the experiment is most useful for investigating how the boundary layer equilibrates.

3) Doubts about microphysics scheme / Microphysics evaluation

In our opinion the criticism on the microphysics scheme used in our LES is not really justified, for the following reasons. In contrast to what the reviewer states, the scheme is actually state-of-the-art. The scheme is double moment, avoids using a temperature-dependent phase function, and treats CCN prognostically. In these aspects the scheme is already much more sophisticated than other LES models used in recent peer-reviewed studies (e.g. Ovchinnikov et al., 2017; Stephens et al., 2018; Zhang et al., 2022).

The main question underlying this comment seems to be what level of sophistication in the microphysics scheme is sufficient for achieving our science goals. This does not necessarily have to be the most complex approach, such as a bin-microphysics scheme, which is hard to constrain with observations and can thus introduce its own unwanted uncertainties. Our results show that the double-moment mixed phase scheme we use reproduces the observed basic cloud structure, mass and phase to a satisfactory degree. That is sufficient to support the studies of entrainment efficiency, which is mainly driven by radiative cooling linked to liquid cloud mass. This justification of the microphysics approach has been added in the revised manuscript.

In principle, we share reviewer's concerns about the saturation adjustment and we originally considered addressing the droplet condensation growth in the discussion. However we deemed it too technical and too detailed for the scope of the article. There are two main reasons for applying the saturation adjustment: 1) consistency and 2) time scale involved. Firstly, the saturation adjustment as applied is part of the scheme (Seifert and Beheng, 2006) and has been used in number of studies of mixed-phase clouds. Secondly, the time steps in our simulations are within the range 0.7–8.1 s. If we consider the droplet growth as described by Lamb and Verlinde (2011) and Pruppacher and Klett (2012), we can estimate the growth speed of cloud droplets in the moist updrafts here. The growth speed implied by the saturation adjustment is within the bounds, and the main factor controlling the size of the droplets is the number of CCN activated. That said. If a simulation examined strongly stratified clouds in tenuous regimes (and thus encountered both short timesteps as well as higher supersaturations due to slower growth rates of larger cloud droplets), then it would be reasonable to test the sensitivity of saturation adjustment vs the explicitly calculated condensational growth of the cloud droplets. However that is not the case here.

Please note that the availability of in-situ observations on both liquid and ice mass, as well as aerosol, is already a big step forward compared to previous LES studies of mixed phase clouds, which often lack one of these datasets or have to completely rely on remote sensing data. That novelty is now mentioned more clearly in the manuscript. The detailed evaluation of cloud droplet and ice concentrations that the reviewer asks for is unfortunately not possible, because the aircraft data is too incidental and sparse for this purpose. For these reasons we decided to evaluate the model in terms of mean (bulk) properties, and not higher order statistics.

4) It is unclear how CCN and IN are prescribed and evolve

In response to this comment we have substantially rewritten the description of the treatment of CCN and IN in the simulation. The time evolution of CCN in the simulation is now also documented.

5) The importance of ice

Cloud ice is important in this case, because a significant part of the total condensate is in frozen state. This applies to suspended (cloud) ice, but also the precipitation in the cloud layer. The ice phase thus plays a key role in the humidity budget of the boundary layer. Similarly, the formation of ice affects the radiative budget, by removing liquid cloud mass which is most important for the longwave radiative cooling that drives the turbulence near the thermal inversion. For these reasons it is important to account for ice in the simulations. We have added this explanation to the manuscript.

6) Technical quality low / Conclusions too short / 'Thin' outcome

Thanks for pointing out the technical shortcomings, this helped us a lot in improving the figures and presentation. The concluding section has also been expanded, and now focuses better on the new insights obtained in relation to the new main hypothesis.

We understand from the reviewer's comment that original sections 6 and 7 were not reviewed. But the key analysis and results were actually presented in exactly these two sections. Accordingly, in our opinion the verdict of "thin outcome" is not supported by the content of the review. Please note that the revised paper has been significantly altered, in particular concerning the main hypothesis and analysis. Accordingly, we hope that these adjustments have made the outcome of the revised manuscript much more substantial.

*Response to detailed comments*

1. Sentence has been rephrased

2. Sensitivity simulations with different CCN and IN concentrations are now included, so that the interactions are properly studied

3. There is value in overlaying pdfs in the plot, it is also a technique that is often used. Information about solubility is added. The main reason for the difference in aerosol above and below the clouds is their removal by precipitation. This is now mentioned, and the evolution of CCN structure is now also documented.

4. The description was corrected. Scalar advection is represented by he centered difference method. We refer to Heus et al. (2010) for the full details.

5. As explained in Section 3.2, the initial CCN concentration is guided directly by the in-situ observations by the P6, as shown in Fig. 4. We do not have measurements of IN concentrations available, for this we follow previous studies. Please note that in using in-situ observations of CCN concentrations we already go far beyond previous LES studies of this kind, which often just use climatology.

6. A description of the long wave radiation is added

7. This statement has been modified, and details have been added to the description.

8. We now consistently use "continuous nudging". However, it should be mentioned at least once that this nudging is Newtonian. The nudging is only applied above the themal inversion height, not below. Indeed the perturbations are simply removed.

9. Yes, the CCN can evolve freely. While no sources of CCN are included (this would be purely speculative), the drop in CCN level during the simulation time is small enough not to matter. This is now mentioned in the text.

10. This type of plots, with a shaded background but no solid axes, has been accepted for publication in numerous journals, and also adheres to the ACP authors' instructions. The grid lines are clearly visible, and the contrast with the white background serves as the panel boundary. The text size in the figure labels has been increased, as requested.

11. As stated above, relaxation is only applied above the thermal inversion, not below it. This method thus prescribes the boundary condition for turbulence, not the turbulence itself. This also means that below the inversion, the agreement between model and measurements is not trivial. We now explain this in more detail in the text.

12. The figure has been removed in the revised manuscript

13. We do not agree, the comparison is actually quite good. With only a few flight legs available, it is much more difficult to quantitatively compare concentrations, compared to "simpler" bulk properties such as mass. For this reason we decided to only evaluation cloud mass and ice, as also explained in our general response. As requested, we further improved the figures. For example, we added the mean observed values, to allow better comparison between model and measurements.

14. The time height figures have been improved, including their description.

15. The word "textbook" has been changed

16-18. These figures have been removed in the revised version.

19. The second part of the results section has been substantially changed.

**Response to Reviewer 2**

*Response to general (major) comments*

We thank the reviewer for the extensive assessment of our manuscript, and for the constructive feedback provided. On the positive side, we are glad that the reviewer agrees that the clouds observed during ACLOUD RF20 are relevant in the context of Arctic climate change, and that this motivates detailed scientific study. Our LES simulations are also considered to be extensive, and the model is judged to be a very capable of reproducing the observed conditions. That said, the reviewer also has some major concerns about the submitted manuscript. To briefly summarize, these issues include:

1. The description and data presentation are not good enough to make results reproducable

2. The description of some aspects of the model and the numerical experiment are not adequate enough

3. A central science question is lacking, and the scientific analysis is limited, in particular concerning aerosol-cloud interactions. Overall the paper does little more than demonstrate that the model's CCN sinks do indeed remove CCN

4. Part II of the result section focuses too much on a single convective event, which might harm representativeness and statistical significance.

5. The analysis does not make enough use of conditional sampling

We find this feedback really helpful. First and foremost, it has made us reconsider the general structure of this study, in particular what type of science question should be addressed and which analysis should be included. The reviewer is right in stating that the simulation described in the manuscript is perhaps most useful to serve as a 'baseline' or control experiment, given the good agreement with the observed thermodynamic and cloudy state. In the revised manuscript, a set of sensitivity experiments is included that deviate from the control setup at crucial points, designed to test a newly formulated hypothesis (as described above in the general response).

Secondly, we agree that the mixing line analysis (part II of the results section) was not substantial enough, and needs more work. We are still of the opinion that the conserved variable diagrams, as well as the other scatter plots, are an efficient way for understanding and visualizing the aerosol-cloud-turbulence interactions in the simulation. However, the application to a single case at a single level is indeed not a quantitative, statistically significant analysis. Accordingly, to give this analysis the space it deserves, we decided to take the mixing line and scatter plot analysis out of this manuscript, and to reserve it for a future publication.

Instead, the sensitivity of the entrainment efficiency to CCN concentrations in mixed-phase convection over open water is now adopted as the central science question of the revised manuscript. This process has been studied for warm clouds (Stevens et al., 2005), but not yet for cold mixed-phase clouds. We find this impact to be significant across the range of CCN concentrations typically observed in Arctic air masses in the area of interest. The efficiency of entrainment also plays an important role in the boundary layer heat budget, as well as the surface energy budget. Both matter for Arctic Amplification and sea ice melt, making this a relevant topic. We investigate in detail how this process works, reporting some unexpected insights.

The adoption of a new main science question and associated analysis make the use of conditional sampling less relevant. The same applies to the time evolution of individual clouds. We agree that these are interesting topics in themselves, but consider them future research topics for now.

Apart from thus rethinking the general structure and content of the paper, we have also made an effort to follow up on the recommendation of improving the figures and descriptions, and bring them up to a scientifically acceptable standard. The title of the manuscript was altered to better reflect the new content. We hope that together, these substantial changes adequately address the reviewer's concerns, and have made the manuscript acceptable for publication in the ACP.

*Response to detailed comments*

Line 30-31: This part of the introduction about cloud classification has been rewritten

Line 48: Phrasing has been altered, as requested

Lines 80-86: Repetitive text has been removed

Line 97-98: Changes applied as requested

Figure 7: The spread is only minimal, which is a non trivial result in itself. So we maintained the percentile shading technique, also because in the next figure, which uses the same technique, the spread is a lot larger. This way, the plotting style is consistent across multiple plots.

Figure 9, and line 328-331: Figure 9 has been removed. Averaging times are now clearly indicated in the other figures.

Figure 11, and line 360: The profiles represent the domain average over a horizontal slice of grid boxes, covering a specified time period. The description has been altered, as requested.

Figure 11, lines 366-367: This figure has been modified, as well as its description, as requested.

...

The detailed comments from "Figure 12" to "Line 525" all refer to figures and text that have been completely removed in the revised manuscript.

...

Line 531: We now refer to the use of reanalysis data as a possible way to establish how typical the stagnant wind conditions occur in the region.

Figure B1: Changes were made as requested.

**References**

Heus, T., van Heerwaarden, C. C., Jonker, H. J., Siebesma, A. P., Axelsen, S., van den Dries, K., ... & de Arellano, J. V. G. (2010). Formulation of and numerical studies with the Dutch Atmospheric Large-Eddy Simulation (DALES). *Geosci. Model Dev*, *3*, 415-444. https://doi.org/10.5194/gmd-3-415-2010

Lamb, D., Verlinde, J. 2011: Physics and chemistry of clouds. Cambridge University Press.

Ovchinnikov et al., 2014: Intercomparison of large-eddy simulations of Arctic mixed-phase clouds: Importance of ice size distribution assumptions, *Journal of Advances in Modeling Earth Systems*, 6, 223–248, https://doi.org/10.1002/2013MS000282

Pruppacher, H.R. and Klett, J.D., 2012: Microphysics of Clouds and Precipitation: *Reprinted 1980*. Springer Science & Business Media.

Seifert, A., Beheng, K., 2006: A two-moment cloud microphysics parameterization for mixed-phase clouds. Part 2: Maritime vs. continental deep convective storms. *Meteorol. Atmos. Phys.* **92,** 67–82. https://doi.org/10.1007/s00703-005-0113-3

Stevens et al., 2005: Evaluation of Large-Eddy Simulations via Observations of Nocturnal Marine Stratocumulus*, Monthly Weather Review*, **133**, 1443 – 1462, https://doi.org/10.1175/MWR2930.1

Stevens et al., 2018: A model intercomparison of CCN-limited tenuous clouds in the high Arctic, *Atmospheric Chemistry and Physics*, 18, https://doi.org/10.5194/acp-18-11041-2018

Zhang et al., 2022: Seasonal cycle of idealized polar clouds: Large eddy simulations driven by a GCM. *Journal of Advances in Modeling Earth Systems*, 14, e2021MS002671. https://doi.org/10.1029/2021MS002671

---

## Referee Report (RR1)

**Review** of a manuscript "Aerosol impacts on the entrainment efficiency of Arctic mixed phase convection in a simulated air mass over open water", previously titled "Aerosol-cloud-turbulence interactions in well-coupled Arctic boundary layers over open water" by Chylik et al.

**Overall recommendation**: publish after substantial (major) revisions

This is a revised manuscript that I reviewed last fall. The paper has been significantly revised and has a new title. With all the changes, the paper should be treated as a new submission, requiring a careful reading from scratch. Overall, I feel the discussion of the revised manuscript should start once again (and be published with comments and responses available online), and not be treated as just a check if the revisions addressed the reviewers' concerns. So either the Editor should reject the original submission and request a new discussion phase, or the authors should withdraw the paper and resubmit the new version for the discussion. In a nutshell, I feel something went wrong, especially considering the length of time between the original submission (November 2021) and submission of the revision (May 2022).

As for the science, the new emphasis provides a better focus of the paper. However, I see significant problems with the technical aspects of the manuscript as presented in my major points and specific line-by-line comments. I have to add that the authors ignored some of my specific comments as a few of those below address the same issues as in the previous review.

**General (major) comments**

1. This is the same as in my previous review: Manuscript title should have "modeling" word in it. Perhaps start with "Modeling of aerosol impacts…".

2. I find the analysis of the assumed initial CCN concentration interesting and worth pursuing. However, I feel the explanation of the simulated dramatic impact is mostly missing. For instance, even without the feedback into dynamics, increasing droplet concentration has to lead to the increase the reflected solar radiation as in the classical Twomey effect. This is not even mentioned in the manuscript. Of course there is a significant dynamical feedback as shown, for instance, by Figs. 11 and 12. The dramatic increase of the LWP feeds back on the Twomey effect, correct? Is that all what happens? I am not sure. For instance, with the low CCN (10 per cc), is there any impact on the precipitation initiation and fallout? In ice-free clouds, low CCN typically implies more drizzle and rain, and thus reduced LWP. How this is modified by the presence of ice? Is the ice initiation in the parameterized microphysics affected by mean droplet size, presumable larger in the low CCN simulation? In a nutshell, I find the new emphasis of the paper interesting, but the analysis of the results leaves much to be desired. This is why I think the paper should go to the open discussion phase again so the authors can openly reply to the reviewers' comments.

3. Fig. 1 in the manuscript is as in the previous version. I commented on it in my previous review and the authors tried to respond. My suggestion is to show a small fraction of the figure, focusing on the area targeted in the LES study. The rest of the figure, difficult to see what it shows, is irrelevant. For instance, where are land boundaries? Another possibility is to add a second panel with the enlarged area targeted by the modeling study to better document cloud systems there.

4. This is following point 6 from my previous review. I still claim that the technical quality of the presentation needs to be improved. Some figures require adjustments to make them legible when included in the printed version, some figures can be improved, some show quantities that are not clearly defined, some captions remain unclear. See specific examples below.

**Specific minor comments**

1. Caption of Fig. 3 is incorrect (see panels b and c, and their description). Please correct.

2. Fig. 4. The caption says "histogram" but the vertical axis says "pdf". So which one is it? Pdf is a histogram with each bin value divided by the bin width. Also, I feel the caption should say that this is the histogram of time during which a given concentration was observed. I feel more details of those observations would be needed. For instance, what was the time resolution of those observations (1sec? 10 sec?).

3. L 144: "We speculate…": Do model results show that aerosol processing is responsible for local fluctuations? If so (as suggested by Fig. 12c), "speculate" is a poor word.

4. Section 3.2, microphysics. It is not clear if all ice variables (cloud ice, snow, graupel) include prediction of both mass and number, that is, as in a complete double moment scheme. It would be appropriate to mention this explicitly.

5. Section 3.3 misses information about the longwave radiation. This was my point 6 from the previous review.

6. Around l. 245 and related to my previous point 8. I am not sure why "Newtonian" is important to say wrt to the relaxation. It would be more important to explicitly state that the relaxation dumps small-scale perturbations, in contrast to the approach when the mean is relaxed to the prescribed profiles. BTW, such relaxation also serves as gravity wave absorber, correct? So why an extra absorber is added in the top 1 km of the domain? Maybe because of a much shorter time scale is used there.

7. L. 258: Considering statistics shown in Fig. 4 from observations, I would like to see similar statistics from model simulations. In particular, what is the range of CCN spatial fluctuations simulated by the model?

8. L. 262: "…using a maximum initial cloud droplet number concentration…". I do not understand what this statement refers to. Please explain.

9. Fig. 5. The most appealing feature of the figure is the sharp maximum of cloud water near the cloud top (also evident in Fig. 8). This is in contrast to warm stratiform clouds that typically feature a uniformly increasing cloud water with height (e.g., Fig. 4 in Stevens et al. 2005 cited by the authors). Is that because of the presence of ice? If one adds ice and water in Fig. 8a, would the total nonprecipitating water follow close-to-linear profile? Also: the variable show is the mixing ratio, not "mass concentration".

10. L. 322: Why does the secondary ice production allow formation of large and heavy ice hydrometeors?

11. Figs. 9 and 10. The figures show box-whiskers data for the observations, and only mean for the model. Can box-whiskers plots be done for the model as well? I appreciate difficulties in comparing a 1D flights with 2D model data, but I feel comparing statistics would make the analysis better. L. 344 states that comparing particle size distributions is impossible, but comparing spatial variability of mass mixing ratios (and maybe concentrations) would make sense.

12. L. 359: Something is missing in this sentence.

13. Figs. 11 and 12 shows the dramatic impact of assumed initial CCN concentration on cloud macro- and microphysics. However, the analysis leaves me wondering where this impact comes from. BTW, liquid cloud fraction is not defined. For instance, is it calculated locally and then averaged horizontally, or is it based on the horizontally-averaged water and ice mixing ratios? L. 385 suggests the latter per "area fraction". Please make it clear.

14. L. 394. How important is the fact that the entrained air is warmer when the boundary layer grows deeper in the 1000 per cc simulation? This aspect is not mentioned in the discussion and I feel it should.

15. Fig. 13 and L. 399: I think "evolutions" would be better than "time series".

16. How is a1 shown in Fig. 13 defined?

17. Figs 14 and 16. Please plot bars side-by-side, not one behind the other. Initially I thought there was some logic in having some bars wider than others, only to notice that they were simply partially hidden. How is $dF_{rad}$ in Fig. 14 exactly defined? How are LW and SW in Fig 16 exactly defined? For instance, are those across the entire boundary layer, or jest near the inversion? The latter drives entrainment, correct?

18. L. 405: "This suggests that ice formations…". Yes, I agree, but it would be nice to see how. This is where the analysis needs to be expanded.

19. L. 421: "Fig. 14f confirms that entrainment rate increases…". I feel additional analysis should explain what physical mechanism(s) are involved.

20. L. 440. One may wonder if the classical Twomey effect is sufficient to explain 200 W m$^{-2}$ difference. I doubt it. The dynamical feedback is likely the key. See major point 2.

21. Since the SST is prescribed, I find the Eq. (3) and its discussion irrelevant. My suggestion is to focus on the dynamical and microphysical effects steaming from the factor 10 CCN concentration change.

22. In view of the above comments, I did not read sections 6 and 7. I expect these will change significantly after all above comments are addressed.

---

## Referee Report (RR2)

This is my last review of this manuscript: I will decline to review this paper again. I strongly feel – as I said in my previous review, included below – that the process worked poorly for this submission. I was not able to find responses to my comments from the previous cycle. I requested major revision and the authors included only cosmetic adjustments to the previous version (modifying a few figures), basically ignoring almost all of my specific comments. My recommendation is to reject this paper and start the discussion phase again so the reviewers can see how the authors responded to their comments. I appreciate that the Editor may disagree with my recommendation and accept the manuscript. Either way is fine with me. I have to add that I am disappointed with the process and will seriously consider any further review invitations from ACP.

I am uploading this comment together with me previous review (from June 2022) below for the Editor convenience.

**Review** of a manuscript "Aerosol impacts on the entrainment efficiency of Arctic mixed phase convection in a simulated air mass over open water", previously titled "Aerosol-cloud-turbulence interactions in well-coupled Arctic boundary layers over open water" by Chylik et al.

**Overall recommendation**: publish after substantial (major) revisions

This is a revised manuscript that I reviewed last fall. The paper has been significantly revised and has a new title. With all the changes, the paper should be treated as a new submission, requiring a careful reading from scratch. Overall, I feel the discussion of the revised manuscript should start once again (and be published with comments and responses available online), and not be treated as just a check if the revisions addressed the reviewers' concerns. So either the Editor should reject the original submission and request a new discussion phase, or the authors should withdraw the paper and resubmit the new version for the discussion. In a nutshell, I feel something went wrong, especially considering the length of time between the original submission (November 2021) and submission of the revision (May 2022).

As for the science, the new emphasis provides a better focus of the paper. However, I see significant problems with the technical aspects of the manuscript as presented in my major points and specific line-by-line comments. I have to add that the authors ignored some of my specific comments as a few of those below address the same issues as in the previous review.

**General (major) comments**

1. This is the same as in my previous review: Manuscript title should have "modeling" word in it. Perhaps start with "Modeling of aerosol impacts…".

2. I find the analysis of the assumed initial CCN concentration interesting and worth pursuing. However, I feel the explanation of the simulated dramatic impact is mostly missing. For instance, even without the feedback into dynamics, increasing droplet concentration has to lead to the increase the reflected solar radiation as in the classical Twomey effect. This is not even mentioned in the manuscript. Of course there is a significant dynamical feedback as shown, for instance, by Figs. 11 and 12. The dramatic increase of the LWP feeds back on the Twomey effect, correct? Is that all what happens? I am not sure. For instance, with the low CCN (10 per cc), is there any impact on the precipitation initiation and fallout? In ice-free clouds, low CCN typically implies more drizzle and rain, and thus reduced LWP. How this is modified by the presence of ice? Is the ice initiation in the parameterized microphysics affected by mean droplet size, presumable larger in the low CCN simulation? In a nutshell, I find the new emphasis of the paper

interesting, but the analysis of the results leaves much to be desired. This is why I think the paper should go to the open discussion phase again so the authors can openly reply to the reviewers' comments.

3. Fig. 1 in the manuscript is as in the previous version. I commented on it in my previous review and the authors tried to respond. My suggestion is to show a small fraction of the figure, focusing on the area targeted in the LES study. The rest of the figure, difficult to see what it shows, is irrelevant. For instance, where are land boundaries? Another possibility is to add a second panel with the enlarged area targeted by the modeling study to better document cloud systems there.

4. This is following point 6 from my previous review. I still claim that the technical quality of the presentation needs to be improved. Some figures require adjustments to make them legible when included in the printed version, some figures can be improved, some show quantities that are not clearly defined, some captions remain unclear. See specific examples below.

**Specific minor comments**

1. Caption of Fig. 3 is incorrect (see panels b and c, and their description). Please correct.

2. Fig. 4. The caption says "histogram" but the vertical axis says "pdf". So which one is it? Pdf is a histogram with each bin value divided by the bin width. Also, I feel the caption should say that this is the histogram of time during which a given concentration was observed. I feel more details of those observations would be needed. For instance, what was the time resolution of those observations (1sec? 10 sec?).

3. L 144: "We speculate…": Do model results show that aerosol processing is responsible for local fluctuations? If so (as suggested by Fig. 12c), "speculate" is a poor word.

4. Section 3.2, microphysics. It is not clear if all ice variables (cloud ice, snow, graupel) include prediction of both mass and number, that is, as in a complete double moment scheme. It would be appropriate to mention this explicitly.

5. Section 3.3 misses information about the longwave radiation. This was my point 6 from the previous review.

6. Around l. 245 and related to my previous point 8. I am not sure why "Newtonian" is important to say wrt to the relaxation. It would be more important to explicitly state that the relaxation dumps small-scale perturbations, in contrast to the approach when the mean is relaxed to the prescribed profiles. BTW, such relaxation also serves as gravity wave absorber, correct? So why an extra absorber is added in the top 1 km of the domain? Maybe because of a much shorter time scale is used there.

7. L. 258: Considering statistics shown in Fig. 4 from observations, I would like to see similar statistics from model simulations. In particular, what is the range of CCN spatial fluctuations simulated by the model?

8. L. 262: "…using a maximum initial cloud droplet number concentration…". I do not understand what this statement refers to. Please explain.

9. Fig. 5. The most appealing feature of the figure is the sharp maximum of cloud water near the cloud top (also evident in Fig. 8). This is in contrast to warm stratiform clouds that typically feature a uniformly increasing cloud water with height (e.g., Fig. 4 in Stevens et al. 2005 cited by the authors). Is that because

of the presence of ice? If one adds ice and water in Fig. 8a, would the total nonprecipitating water follow close-to-linear profile? Also: the variable show is the mixing ratio, not "mass concentration".

10. L. 322: Why does the secondary ice production allow formation of large and heavy ice hydrometeors?

11. Figs. 9 and 10. The figures show box-whiskers data for the observations, and only mean for the model. Can box-whiskers plots be done for the model as well? I appreciate difficulties in comparing a 1D flights with 2D model data, but I feel comparing statistics would make the analysis better. L. 344 states that comparing particle size distributions is impossible, but comparing spatial variability of mass mixing ratios (and maybe concentrations) would make sense.

12. L. 359: Something is missing in this sentence.

13. Figs. 11 and 12 shows the dramatic impact of assumed initial CCN concentration on cloud macro- and microphysics. However, the analysis leaves me wondering where this impact comes from. BTW, liquid cloud fraction is not defined. For instance, is it calculated locally and then averaged horizontally, or is it based on the horizontally-averaged water and ice mixing ratios? L. 385 suggests the latter per "area fraction". Please make it clear.

14. L. 394. How important is the fact that the entrained air is warmer when the boundary layer grows deeper in the 1000 per cc simulation? This aspect is not mentioned in the discussion and I feel it should.

15. Fig. 13 and L. 399: I think "evolutions" would be better than "time series".

16. How is a1 shown in Fig. 13 defined?

17. Figs 14 and 16. Please plot bars side-by-side, not one behind the other. Initially I thought there was some logic in having some bars wider than others, only to notice that they were simply partially hidden. How is $dF_{rad}$ in Fig. 14 exactly defined? How are LW and SW in Fig 16 exactly defined? For instance, are those across the entire boundary layer, or just near the inversion? The latter drives entrainment, correct?

18. L. 405: "This suggests that ice formations…". Yes, I agree, but it would be nice to see how. This is where the analysis needs to be expanded.

19. L. 421: "Fig. 14f confirms that entrainment rate increases…". I feel additional analysis should explain what physical mechanism(s) are involved.

20. L. 440. One may wonder if the classical Twomey effect is sufficient to explain 200 W m$^{-2}$ difference. I doubt it. The dynamical feedback is likely the key. See major point 2.

21. Since the SST is prescribed, I find the Eq. (3) and its discussion irrelevant. My suggestion is to focus on the dynamical and microphysical effects steaming from the factor 10 CCN concentration change.

22. In view of the above comments, I did not read sections 6 and 7. I expect these will change significantly after all above comments are addressed.

---

## Author Response (AR2)

**Response to Reviewers**

**Response to anonymous reviewer 1**

Review of a manuscript "Aerosol impacts on the entrainment efficiency of Arctic mixed phase convection in a simulated air mass over open water", previously titled "Aerosol-cloud-turbulence interactions in well- coupled Arctic boundary layers over open water" by Chylik et al.

Overall recommendation: publish after substantial (major) revisions

This is a revised manuscript that I reviewed last fall. The paper has been significantly revised and has a new title. With all the changes, the paper should be treated as a new submission, requiring a careful reading from scratch. Overall, I feel the discussion of the revised manuscript should start once again (and be published with comments and responses available online), and not be treated as just a check if the revisions addressed the reviewers' concerns. So either the Editor should reject the original submission and request a new discussion phase, or the authors should withdraw the paper and resubmit the new version for the discussion. In a nutshell, I feel something went wrong, especially considering the length of time between the original submission (November 2021) and submission of the revision (May 2022).

We thank the reviewer for the careful assessment of our revised manuscript. We were surprised to read the assessment that "something went wrong", and about the advice to treat the revised paper as a new submission. In our opinion this criticism is unjustified, for various reasons.

Firstly, we followed the revision instructions from the editorial office to the letter, including the invitation to submit a revised version (and not a new submission). We also obeyed the deadline for submission, so the length of time can not be considered overly lengthy. Accordingly, we feel that the comments on the followed procedure should not be part of the response to us authors, but should be aimed at the editorial office instead.

Secondly, while the modifications in the revised manuscript are indeed substantial, this is not unusual for a major revision. In my experience in submitting and reviewing papers I have seen much more drastic changes under the label 'major revision', including changes in the title. And the first (and largest) part of the paper describing the case, the LES experiments and their evaluation against observations, has seen relatively little change. Accordingly, in our opinion (which is apparently shared by the editor) the verdict "reject" would be

overly harsh.

As for the science, the new emphasis provides a better focus of the paper. However, I see significant problems with the technical aspects of the manuscript as presented in my major points and specific line- by-line comments. I have to add that the authors ignored some of my specific comments as a few of those below address the same issues as in the previous review.

We tried hard to follow up on the original recommendation to improve the technical aspects of the paper. We think this has substantially improved the manuscript, in particular some of the figures. We apologize if we omitted some of these recommendations in our first revision; we hope that in the second revision we have adequately addressed these points.

**General (major) comments**

1. This is the same as in my previous review: Manuscript title should have "modeling" word in it. Perhaps start with "Modeling of aerosol impacts...".

The title now has the word "simulated" in it. "Simulating" is a subset of "modeling", which can include many other types of numerical integrations (such as global circulation modeling). So we did follow up on this comment. We think "simulated" is more appropriate in our title, because i) it is more precise and accurate and thus delineates better from other atmospheric modeling studies, and ii) we hereby follow many other previous LES studies that have the word "simulation" (and not modeling) in the title.

2. I find the analysis of the assumed initial CCN concentration interesting and worth pursuing. However, I feel the explanation of the simulated dramatic impact is mostly missing. For instance, even without the feedback into dynamics, increasing droplet concentration has to lead to the increase the reflected solar radiation as in the classical Twomey effect. This is not even mentioned in the manuscript. Of course there is a significant dynamical feedback as shown, for instance, by Figs. 11 and 12. The dramatic increase of the LWP feeds back on the Twomey effect, correct? Is that all what happens? I am not sure. For instance, with the low CCN (10 per cc), is there any impact on the precipitation initiation and fallout? In ice-free clouds, low CCN typically implies more drizzle and rain, and thus reduced LWP. How this is modified by the presence of ice? Is the ice initiation in the parameterized microphysics affected by mean droplet size, presumable larger in the low CCN simulation? In a nutshell, I find the new emphasis of the paper interesting, but the analysis of the results leaves much to be desired. This is why I think the paper should go to the open discussion phase again so the authors can openly reply to the reviewers' comments.

We fully agree that increasing droplet concentration leads to the increase in reflected solar radiation, and the dramatic increase of the LWP is likely to feed back on this effect. As suggested, we have added the estimation of the Twomey effect. We have incorporated the this effect directly after the paragraph on 2nd cloud-aerosol indirect effect, where it fit thematically. Our comparison shows

that there is a good agreement for the pristine case, however there are significant differences for the control case and the continental case.

Regarding the impact of CCN concentrations the ice processes, we have added further explanation of the effect of the cloud droplet size, as well as figure showing the tendencies in the mass of ice hydrometeors

3. Fig. 1 in the manuscript is as in the previous version. I commented on it in my previous review and the authors tried to respond. My suggestion is to show a small fraction of the figure, focusing on the area targeted in the LES study. The rest of the figure, difficult to see what it shows, is irrelevant. For instance, where are land boundaries? Another possibility is to add a second panel with the enlarged area targeted by the modeling study to better document cloud systems there.

In the new version we follow both suggestions: the main panel now focuses on the target area, while the smaller secondary panel (a) show the general location of the mission with the land boundaries clearly marked. We agree that this configuration of the figure is more efficient in describing the flights and the simulated domain.

4. This is following point 6 from my previous review. I still claim that the technical quality of the presentation needs to be improved. Some figures require adjustments to make them legible when included in the printed version, some figures can be improved, some show quantities that are not clearly defined, some captions remain unclear. See specific examples below.

Thanks for this critical but constructive comment. We have again looked at all figures, and accommodated all suggested improvements as provided below. We hope that by doing so the figures now have the scientific quality required for publication.

**Specific minor comments**

1. Caption of Fig. 3 is incorrect (see panels b and c, and their description). Please correct.

The caption has been corrected to agree with the panels.

2. Fig. 4. The caption says "histogram" but the vertical axis says "pdf". So which one is it? Pdf is a histogram with each bin value divided by the bin width. Also, I feel the caption should say that this is the histogram of time during which a given concentration was observed. I feel more details of those observations would be needed. For instance, what was the time resolution of those observations (1sec? 10 sec?).

The caption of the figure was indeed misleading. We have corrected it now to "pdf". We have also added some of the main properties of the observations

and for further information we refer the data publications.

3. L 144: "We speculate. . . ": Do model results show that aerosol processing is responsible for local fluctuations? If so (as suggested by Fig. 12c), "speculate" is a poor word.

Replaced with the word "hypothesize"

4. Section 3.2, microphysics. It is not clear if all ice variables (cloud ice, snow, graupel) include prediction of both mass and number, that is, as in a complete double moment scheme. It would be appropriate to mention this explicitly.

Adding explanatory sentence: This is a full 2-moment implementation, the mass concentration and number concentration of each of five hydrometeors are thus prognostic variables.

5. Section 3.3 misses information about the longwave radiation. This was my point 6 from the previous review.

This was an unfortunate oversight on our side. The composite large-scale state above the model ceiling is indeed applied also for the downward longwave radiative flux. We have modified the text accordingly. For further information on long wave and short wave radiation, readers are referred in the section 3.1 to Pincus & Stevens (2009).

6. Around l. 245 and related to my previous point 8. I am not sure why "Newtonian" is important to say wrt to the relaxation. It would be more important to explicitly state that the relaxation dumps small-scale perturbations, in contrast to the approach when the mean is relaxed to the prescribed profiles. BTW, such relaxation also serves as gravity wave absorber, correct? So why an extra absorber is added in the top 1 km of the domain? Maybe because of a much shorter time scale is used there.

We explicitly state that the continuous nudging applied is Newtonian relaxation as opposed and not some more complicated spectral nudging. The continuous nudging here is towards the prescribed profiles with the purpose of preventing excessive model drift in this height range, and reducing perturbations in the temporal domain. At each vertical level, the difference between the prescribed profile and the current horizontal mean is added to all gridpoints (Neggers et al., 2012), independent of how far from the horizontal mean they are. Therefore, it neither acts as gravity wave absorber nor removes perturbations small-scale spatial perturbations.

The sponge layer instead nudge values at each gridpoint towards the horizontal mean. The strength of this nudging is increasing with altitude. At the top of domain it reaches the timescale of $\approx 6$ minutes (Heus et al., 2010).

For clarity we have added references to sources that describe continuous nudging and sponge layer.

7. L. 258: Considering statistics shown in Fig. 4 from observations, I would like to see similar statistics from model simulations. In particular, what is the range of CCN spatial fluctuations simulated by the model?

The full 3D fields of CCN concentration were unfortunately not amongst the stored outputs of the simulations. That said, such comparison would be statistically inconsistent: the LES follows the trajectory of the air mass, while the aircraft sampled on a race track which can be also affected by mesoscale variability.

8. L. 262: "...using a maximum initial cloud droplet number concentration...". I do not understand what this statement refers to. Please explain.

Replaced with brief description and explanation:
The initial cloud droplet number concentration is supersaturated areas is set accordingly: the initial mean droplet size must be higher than the threshold $\bar{x}_{min} = 4.2 \cdot 10^{-12}$ g used by Seifert & Beheng (2006), and at most 1/2 of the initial CCN number concentration is activated. The motivation for this initial concentration is straightforward: the simulation can starts with neither too large nor too small droplets, and spins-up without encountering lack of CCN.

9. Fig. 5. The most appealing feature of the figure is the sharp maximum of cloud water near the cloud top (also evident in Fig. 8). This is in contrast to warm stratiform clouds that typically feature a uniformly increasing cloud water with height (e.g., Fig. 4 in Stevens et al. 2005 cited by the authors). Is that because of the presence of ice? If one adds ice and water in Fig. 8a, would the total nonprecipitating water follow close-to-linear profile? Also: the variable show is the mixing ratio, not "mass concentration".

We understand the maximum near the cloud top seems to be sharp, however it can be explained by the effect of ice microphysics: the ice precipitation is removing cloud water from the area below the maxima. This can be clearly seen in the Figure 8, which shows the precipitation in the panel b and the suspended cloud liquid water and ice water in the panel b. We can clearly see a match between the gradient of cloud water mass around 800–1300 m and the mass of ice hydrometeors. We can also see relative change in the mases of snow to graupel, indicationg the riming of cloud droplets (showing the riming tendencies and other hydrometeor tendencies was considered beyond the scope of this paper).

With regard to cited literature The Figure. 4 in Stevens et al. (2005), we also have to consider the spatial scales. Firstly, the distance between the cloud top and the cloud bottom there is there only about 250 m. Both the gradient below and the gradient above are sharper than in our simulations (as seen in Figure 5 and ore clearly in Figure 8).

We have corrected the caption of figure 5 from "mixing ratio" to "specific mass".

10. L. 322: Why does the secondary ice production allow formation of large

and heavy ice hydrometeors?

The secondary ice production in supercooled liquid clouds leads to high number of rimed ice hydrometeors (and partially rimed ice hydrometeors) as well as to spread in sizes of ice hydrometeors, and thus also their terminal velocities. Due to differences in the terminal velocities of ice particles they are more likely to further aggregate. In the text we added reference to Sullivan et al. (2017) and Georgakaki et al.(2022) that both deal with this phenomenon.

11. Figs. 9 and 10. The figures show box-whiskers data for the observations, and only mean for the model. Can box-whiskers plots be done for the model as well? I appreciate difficulties in comparing a 1D flights with 2D model data, but I feel comparing statistics would make the analysis better. L. 344 states that comparing particle size distributions is impossible, but comparing spatial variability of mass mixing ratios (and maybe concentrations) would make sense.

We had also consider this comparison, but we have realised that it is better not to. The main reason is that the whiskers would then be statistically inconsistent: the sample size of the LES data is much larger compared to the observations. This is already mentioned in the text. To avoid raising the wrong expectations, we choose here to only show the 1st statistical moment, the mean.

12. L. 359: Something is missing in this sentence.

We have added missing preposition. The sentence is now
"by the strong variability in $w$ in the close vicinity of convective cells."

13. Figs. 11 and 12 shows the dramatic impact of assumed initial CCN concentration on cloud macro- and microphysics. However, the analysis leaves me wondering where this impact comes from. BTW, liquid cloud fraction is not defined. For instance, is it calculated locally and then averaged horizontally, or is it based on the horizontally-averaged water and ice mixing ratios? L. 385 suggests the latter per "area fraction". Please make it clear.

The *liquid cloud fraction* is caculated locally. It is defined as the fraction of the gridpoints at each vertical layer where the cloud liquid water content is above the threshold (set to $0.01\,\mathrm{g\,m^{-2}}$ for cloud liquid water).

We have added the description to the caption of the figure.

14. L. 394. How important is the fact that the entrained air is warmer when the boundary layer grows deeper in the 1000 per cc simulation? This aspect is not mentioned in the discussion and I feel it should.

It is indeed an imortant point. We have added the following sentence:"The strengthening can be explained as a consequence of a deep mixed layer growing into a weakly stable overlying layer."

15. Fig. 13 and L. 399: I think "evolutions" would be better than "time

series".

adjusted as proposed

16. How is a1 shown in Fig. 13 defined?

The following definition of the model output has been added to the manuscript:

The liquid cloud cover $a_l$ is in the model defined as the fraction of the domain where the vertically integrated cloud liquid water exceeds the minimum threshold of $0.01\,\mathrm{g\,m^{-2}}$.

17. Figs 14 and 16. Please plot bars side-by-side, not one behind the other. Initially I thought there was some logic in having some bars wider than others, only to notice that they were simply partially hidden. How is dF rad in Fig. 14 exactly defined? How are LW and SW in Fig 16 exactly defined? For instance, are those across the entire boundary layer, or just near the inversion? The latter drives entrainment, correct?

We have adjusted the bars in order so they are not overlaying each ther anymore.

The $dF_{\mathrm{rad}}$ was a typo in the notation. The correct is $\Delta F_{\mathrm{rad}}$, the net long wave radiative flux divergence across the liquid cloud layer. We have corrected the notation.

The values of radiative components in the figure 17a (previously 16a) explicitly refer to radiative surface fluxes (as stated in the caption), horizontally averaged over the whole surface of the domain (this is now added to the caption).

18. L. 405: "This suggests that ice formations. . . ". Yes, I agree, but it would be nice to see how. This is where the analysis needs to be expanded.

A few explanatory sentences were added before this statement, as well as Fig. 15 showing the tendencies in ice water budget.

19. L. 421: "Fig. 14f confirms that entrainment rate increases. . . ". I feel additional analysis should explain what physical mechanism(s) are involved.

The phenomenon is briefly described five lines earlier, the entrainment rate is further investigated in the following paragraphs, and the consequences further discussed in (6.3).

20. L. 440. One may wonder if the classical Twomey effect is sufficient to explain 200 W m -2 difference. I doubt it. The dynamical feedback is likely the key. See major point 2.

The Twomey effect would suggest even larger difference than 200 W m$^{-2}$ due to increase in both the number of cloud droplets, as well as due to dramatic

increase in Liquid water path.

This result is now desribed earlier in the manuscript, and here we are adding just a brief reference to the figure.

21. Since the SST is prescribed, I find the Eq. (3) and its discussion irrelevant. My suggestion is to focus on the dynamical and microphysical effects steaming from the factor 10 CCN concentration change.

We do not agree. Equation (3) does not define the SST, but the Surface Energy Budget (SEB) that consists of energy fluxes, which is something entirely different. While the SST is indeed fixed the CCN change does significantly affect the SEB, in ways not yet reported in previous studies. Because the SEB plays a crucial role in Arctic climate change, in particular sea ice melt in response to cloud cover, we feel this insight is relevant and should be reported.

22. In view of the above comments, I did not read sections 6 and 7. I expect these will change significantly after all above comments are addressed.

**Response to reviewer 2**

The authors have done an excellent job in addressing my concerns with the original manuscript. The new material on the impact of CCN concentration on entrainment is novel and likely to be of wider interest. Entrainment is an issue that is difficult to represent well in global models anywhere, and little studied in the Arctic. A topic well worth pursuing.

I am happy to recommend publication with only minor technical revisions for the points noted below.

We thank the reviewer for the careful assessment of our revised manuscript and the encouraging words. In the following paragraphs we will address all the minor comments.

line 258: "as shown in 4" as shown in Figure 4"

Corrected as suggested.

line 325: 'for both instruments' → 'for each instrument'

Corrected as suggested.

line 389 - 'the stratiform cloud layer shows signs of decoupling...' - it would be useful here to explain what these signs of decoupling are. I assume the sudden drop in altitude at 40-50 hours, in the dotted line indicating 'BL' top in figure 11 is the primary indicator, along with the increase in cloud fraction below this. Is there any indication that this might be linked to changes in turbulence generated at cloud top resulting from the diurnal cycle is solar radiation?

The signs of decoupling can be seen in virtual potential profiles for these times (not shown) and a gap in cloud fraction forming above 1100 m, nearly separating the cloud layer into two distinct layers in the interval between 32 and 48 hour(Figure 5.c).

We have added this explanation to the text.

line 433 (and elsewhere) - 'larger CCN' should really be 'larger CCN concentration' to avoid the implication of larger CCN particle size. It's implicit from context, but jars a little when reading.

Modiefied. Based on the context, we replaced "lager CCN" with "higher CCN concentrations", and "larger CCN range" with "larger CCN concentration range".

line 435: 'Apparently, the boundary layer responds...strengthening the inversion...' - worth noting that the strengthening of the inversion is a simple consequence of a deeper mixed layer, grown into a weakly stable overlying layer.

We agree that the strengthening of the inversion is mostly a consequence of a deeper mixed layer growing into a weakly stable overlying layer. We have added an explanatory sentence between " thermal inversion strength (shown in Figure 14h)" and "Apparently, ... "

line 508: 'the amount of energy lost to deeper ocean layers' - it's not clear if the authors simply mean deeper within the ocean mixed layer, or energy transferred across the thermocline/pycnoline into the deep ocean. The latter is unlikely to be significant except in the case of deep water formation events in winter. The cumulative impact on the temperature of the mixed layer might, however, impact on deep water formation via the amount of cooling needed to trigger it.

We meant the energy lost to the deep layers of the ocean, however we agree that the energy transferred across the thermocline is unlikely during this season. Therefore we are adjusting this sentence accordingly "energy lost to deeper parts of the oceanic mixed layer".